# OMNIPHYSGS: 3D CONSTITUTIVE GAUSSIANS FOR GENERAL PHYSICS-BASED DYNAMICS GENERATION

**Yuchen Lin**[1]**, Chenguo Lin**[†1]**, Jianjin Xu**[2]**, Yadong Mu**[‡1]
[1]Peking University, [2]Carnegie Mellon University
**https://wgsxm.github.io/projects/omniphysgs**

## ABSTRACT

Recently, significant advancements have been made in the reconstruction and generation of 3D assets, including static cases and those with physical interactions. To recover the physical properties of 3D assets, existing methods typically assume that all materials belong to a specific predefined category (*e.g.*, elasticity). However, such assumptions ignore the complex composition of multiple heterogeneous objects in real scenarios and tend to render less physically plausible animation given a wider range of objects. We propose OMNIPHYSGS for synthesizing a physics-based 3D dynamic scene composed of more general objects. A key design of OMNIPHYSGS is treating each 3D asset as a collection of constitutive 3D Gaussians. For each Gaussian, its physical material is represented by an ensemble of 12 physical domain-expert sub-models (rubber, metal, honey, water, etc.), which greatly enhances the flexibility of the proposed model. In the implementation, we define a scene by user-specified prompts and supervise the estimation of material weighting factors via a pretrained video diffusion model. Comprehensive experiments demonstrate that OMNIPHYSGS achieves more general and realistic physical dynamics across a broader spectrum of materials, including elastic, viscoelastic, plastic, and fluid substances, as well as interactions between different materials. Our method surpasses existing methods by approximately 3% to 16% in metrics of visual quality and text alignment.

## 1 INTRODUCTION

Synthesizing realistic 3D dynamic (*i.e.*, 4D) scenes has emerged as an attractive task with the development of 3D reconstruction and video generation techniques. Recent advances in 3D differentiable rendering (Mildenhall et al., 2020; Kerbl et al., 2023) establish effective grounds for learning-based dynamic scene synthesis. Although some methods (Ren et al., 2023; Zhao et al., 2023; Yin et al., 2023b; Jiang et al., 2024b; Ling et al., 2024) have presented vivid 3D dynamics, the non-physical nature of data-driven approaches inevitably leads to artifacts and inconsistencies since the learned models are not strictly constrained by the physical laws governing the real world. To this end, we advocate that the incorporation of physical priors is essential for 4D scene synthesis, which guides the learning process to generate more realistic and physically plausible results.

In this work, we choose to use 3D Gaussian Splatting (3DGS) (Kerbl et al., 2023) to represent the scene due to its particle nature, allowing the assignment of physical properties to each Gaussian particle. Related works (Zhong et al., 2024; Fu et al., 2024; Lin et al., 2024; Qiu et al., 2024; Borycki et al., 2024; Feng et al., 2024) have investigated the integration of physical priors in the context of Gaussian Splatting. PhysGaussian (Xie et al., 2024) initially introduces the Material Particle Method (MPM) (Stomakhin et al., 2013; Jiang et al., 2016) to simulate the dynamics of 3DGS. Subsequent studies (Zhang et al., 2024b; Liu et al., 2024; Huang et al., 2024) extend MPM to learning-based dynamic 3D scene synthesis by modeling physical properties with video supervision (Poole et al., 2023). However, some methods require manual tuning of physical properties (Xie et al., 2024), which is time-consuming and may lead to suboptimal results, as estimating material properties requires expert knowledge. Other methods are limited to specific material types, such as pure elastic materials (Zhang et al., 2024b; Huang et al., 2024) or viscoelastic materials (Liu et al., 2024), but do not support multiple materials within a single scene. As a result, they fail to offer a general and automatic solution for dynamic scene synthesis.

---

†: Project lead; ‡: Corresponding author.

Figure 1: **Comparison with Previous Methods**. Existing methods rely on handcrafted or narrowly restrictive physical models (*e.g.*, pure elasticity) that limit generalizability. Our method introduces Constitutive Gaussians to better represent physical materials, thus achieving more automatic and physically plausible dynamic synthesis of various materials within a unified framework.

Our task is to synthesize general physics-based 3D dynamics, where *general dynamics* is defined as generating dynamics for a wide range of materials, including pure elastic, viscoelastic, plastic, and fluid substances, as well as interactions between different materials within a single scene. To achieve this, ideal physical guidance should (1) provide a strong physical constraint for the dynamics, (2) be flexible in handling various kinds of materials and objects, and (3) be capable of learning without the burden of manual modeling. Constitutive models (Gonzalez & Stuart, 2008; Jiang et al., 2016) are physical models that describe the material responses to different mechanical conditions, providing the relationship between two physical quantities such as stress and strain. These models, which are typically derived from experts' observations of material mechanisms, are decisive factors in determining the dynamics of different materials. To enhance the flexibility and automation of dynamic synthesis, it's crucial to generalize the constitutive model instead of manually setting it as previous works (Xie et al., 2024; Zhang et al., 2024b; Liu et al., 2024; Huang et al., 2024) did.

In light of this, we propose OMNIPHYSGS as a novel framework for general physics-based 3D dynamic synthesis. Our method starts with obtaining static Gaussian particles from multi-view images, and we adopt MPM to simulate the dynamics of 3DGS. To generalize the representation of physical materials, we extend vanilla Gaussians to the proposed **Constitutive Gaussians** by introducing a learnable constitutive model for each Gaussian particle. Constitutive Gaussians are designed as a physics-guided neural network, where features of Gaussian kernels are extracted and processed by a physical-aware decoder to predict strain and stress. These predictions are incorporated with a set of expert-designed constitutive models to handle various materials and enhance the physical plausibility of dynamic scenes. The simulated states of Constitutive Gaussians are rendered into video frames and passed to a pre-trained video diffusion model with a text prompt. Finally, learnable Constitutive Gaussians are optimized with the Score Distillation Sampling (SDS) (Poole et al., 2023).

To the best of our knowledge, OMNIPHYSGS is the first method to introduce learnable constitutive models for physics-based 3D dynamic synthesis (Xie et al., 2024; Zhang et al., 2024b; Huang et al., 2024; Liu et al., 2024). As shown in Figure 1, the core contribution of our method, **general physics-based 3D dynamic synthesis**, refers to the automatic synthesis of dynamics and interactions between heterogeneous materials, all while maintaining physical plausibility. This is achieved by leveraging MPM's physical constraints, the generalizability of Constitutive Gaussians, and the

material knowledge encoded in the video diffusion model. Extensive experiments demonstrate that OMNIPHYSGS can generate physically plausible motions and interactions of various materials, without any manual tuning of physical properties. Comprehensive comparisons with state-of-the-art methods and ablation studies validate the effectiveness of each component of our method.

## 2 RELATED WORK

**4D Generation** Most efforts in 4D generation leverage text-to-image and video diffusion models to distill 4D objects using SDS (Poole et al., 2023), employing various dynamic 3D representations, such as HexPlane (Singer et al., 2023), multi-scale 4D grids (Zhao et al., 2023), K-plane (Jiang et al., 2024b), multi-resolution hash encoding (Bahmani et al., 2024), disentangled canonical NeRF (Zheng et al., 2024) or 3D Gaussians (Ling et al., 2024) with a deformation field, and warped Gaussian surfels (Wang et al., 2024a). To facilitate multi-view spatial-temporal consistency modeling, recent works (Zhang et al., 2024a; Jiang et al., 2024a) concentrate on multi-view video generative models to provide enhanced gradients for distillation. In addition to SDS supervision, other studies propose generating videos first as direct references for appearance and motion to optimize 3D representations (Ren et al., 2023; Yin et al., 2023b; Pan et al., 2024; Zeng et al., 2024) or utilizing a generalizable reconstruction model to avoid the time-consuming distillation process (Ren et al., 2024). However, none of these approaches guarantee the physical fidelity of generated 4D contents, due to the lack of physical constraints during optimization.

**Interactive Dynamics Generation** Existing works have investigated interactive dynamics generation for both 2D and 3D content with respect to user preferences or constraints. For image animation, various initial conditions have been utilized to guide the generation process, such as driving videos (Siarohin et al., 2019a;b; 2021; Karras et al., 2023), keypoint trajectories (Hao et al., 2018; Blattmann et al., 2021; Chen et al., 2023a; Yin et al., 2023a; Li et al., 2024), or text prompts (Ho et al., 2022; Yang et al., 2023; Chen et al., 2023b;c; Zhang et al., 2023). Recent works (Jiang et al., 2024a; Ling et al., 2024) extend the interactive dynamics generation to 3D content. To ensure the physical plausibility of the generated dynamics, a series of works (Li et al., 2023; Qiu et al., 2024; Borycki et al., 2024; Zhong et al., 2024; Fu et al., 2024; Feng et al., 2024) have proposed to introduce physical constraints. Among these, PhysGaussian (Xie et al., 2024) married a physical simulation method, MPM, with Gaussian representations. However, since current 3D representations are not naturally endowed with material properties, PhysGaussian requires manually specifying material properties for each particle. Subsequent works (Huang et al., 2024; Zhang et al., 2024b; Liu et al., 2024) attempt to learn physical parameters from diffusion models but are restricted to specific types of materials, such as elasticity. In contrast, our method aims to synthesize 3D dynamics with physical fidelity for various materials and complex object combinations.

## 3 METHOD

### 3.1 PROBLEM STATEMENT

3D Gaussian Splatting (Kerbl et al., 2023) is employed to represent 3D scenes, offering a good trade-off between simplicity and expressivity. A set of 3D Gaussian kernels $\{\mathbf{x}_p, \boldsymbol{\Sigma}_p, \mathrm{sh}_p, \alpha_p\}_{p \in \mathcal{P}}$ is used as a representation, where $\mathbf{x}_p \in \mathbb{R}^3$ is the center of the kernel, $\boldsymbol{\Sigma}_p$ is its covariance matrix, $\mathrm{sh}_p$ represents spherical harmonic coefficients, and $\alpha_p$ is the kernel opacity. These kernels can be splatted onto 2D image planes for different cameras to render a 3D scene from arbitrary viewpoints.

We are addressing the task of synthesizing physically plausible 3D dynamics for heterogeneous objects. Given a 3D scene $\mathcal{S}$ represented by a set of 3D Gaussian kernels $\{\mathbf{x}_p, \boldsymbol{\Sigma}_p, \mathrm{sh}_p, \alpha_p\}_{p \in \mathcal{P}}$ and a text prompt $\mathbf{y}$ describing the motion or material of different objects in the scene, our goal is to synthesize the 3D dynamics that align with the prompt and simultaneously adhere to physical laws.

### 3.2 OMNIPHYSGS

As shown in Figure 2, our method simulates the dynamic scene with the Material Point Method and renders a video clip $\mathcal{V} = [I^{t_0}, I^{t_1}, \cdots, I^{t_{N-1}}]$ via the 3DGS renderer, which is then optimized by a pre-trained text-to-video diffusion model $\Phi$ using Score Distillation Sampling (Poole et al., 2023).

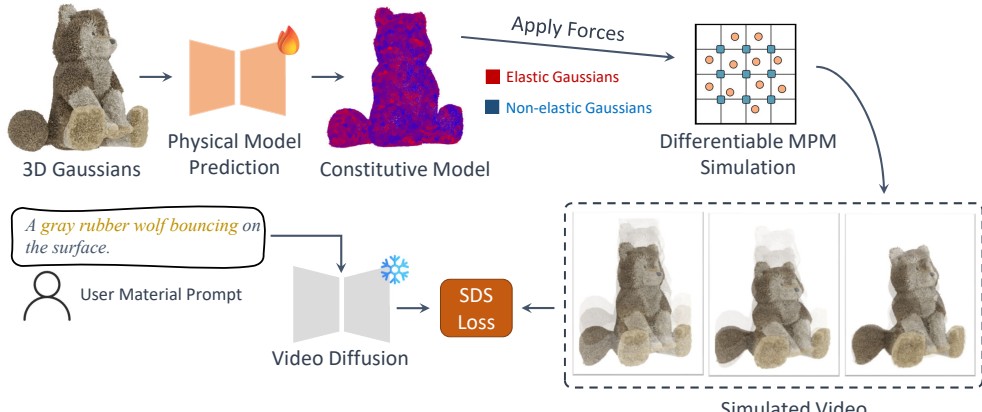

Figure 2: **Method Overview**. OMNIPHYSGS extends 3D Gaussians with learnable constitutive models, introducing Constitutive Gaussians to the differentiable Material Point Method (MPM). A pre-trained video diffusion model is used to guide the training with Score Distillation Sampling.

During the training process, we propose **Constitutive Gaussians** that are trained to predict the optimal material properties of each Gaussian particle, enabling an automatic simulation of general 3D dynamics that align with given text prompts. $\theta$ denotes the learnable parameters of Constitutive Gaussians, which can vary for objects in different materials within the same scene as exemplified in Figure 1. Our objective is to find the optimal parameters $\theta^*$ that maximize the log-likelihood of the distribution modeled by the large pre-trained text-to-video models $\Phi$ as:

$$\theta^* = \arg\max_{\theta} \ \log P\left(\mathcal{V}(\theta)|\mathcal{S}, \mathbf{y}; \Phi\right). \tag{1}$$

### 3.2.1 CONSTITUTIVE GAUSSIAN

Constitutive models are expert-designed functions characterizing how materials deform under specific mechanical conditions. These models are widely adopted in the field of continuum mechanics (Gonzalez & Stuart, 2008; Jiang et al., 2016) to solve the conservation equations of mass and momentum:

$$\frac{\mathrm{D}\rho}{\mathrm{D}t} + \rho\nabla \cdot \mathbf{v} = 0, \quad \rho\frac{\mathrm{D}\mathbf{v}}{\mathrm{D}t} = \nabla \cdot \boldsymbol{\sigma} + \mathbf{f}^{\mathrm{ext}}, \tag{2}$$

where

$$\boldsymbol{\sigma}(\mathbf{x}, t) = \frac{1}{\det(\mathbf{F})}\frac{\partial\Psi}{\partial\mathbf{F}}(\mathbf{F}^E)\mathbf{F}^{ET} \in \mathbb{R}^{3\times3}, \tag{3}$$

represents the Cauchy stress tensor. $\Psi : \mathbb{R}^{3\times3} \to \mathbb{R}^{3\times3}$ is the hyperelastic energy density function and $\mathbf{F}^E \in \mathbb{R}^{3\times3}$ denotes the elastic part of the deformation gradient $\mathbf{F} \in \mathbb{R}^{3\times3}$. In practice, the elastic part of the deformation gradient needs to be corrected as $\mathbf{F} = \psi(\mathbf{F}^E)$, where $\psi : \mathbb{R}^{3\times3} \to \mathbb{R}^{3\times3}$ is a return function that applies plasticity constraints to $\mathbf{F}^E$. Both $\Psi$ and $\psi$ are referred to as **constitutive models**, which describe the material behavior.

In the Material Point Method (see Algorithm 1 in Appendix A.2), the only factors that determine the material properties of each particle are constitutive models $\Psi(\cdot)$ and $\psi(\cdot)$ and the physical parameters $\boldsymbol{\gamma} \in \mathbb{R}^K$, in functions F2Stress and ReturnMap, with the rest of the MPM pipeline being independent of the material properties. To this end, we propose to extend vanilla Gaussian kernels to **Constitutive Gaussian** kernels, generalizing the learning process of material properties in the MPM simulation. We introduce a learnable constitutive model for each Constitutive Gaussian particle as:

$$\Psi \leftarrow \Psi_{\theta_{el}}, \quad \psi \leftarrow \psi_{\theta_{pl}}, \quad \boldsymbol{\gamma} \leftarrow \boldsymbol{\gamma}_{\theta_{phy}}, \tag{4}$$

where $\theta_{el}$, $\theta_{pl}$, and $\theta_{phy}$ are the learnable parameters of neural networks that represent the hyperelastic energy density function, the plasticity return function, and the physical parameters of the material, respectively. Therefore, the training target can be formulated as follows:

$$\min_{\theta_{el}, \theta_{pl}, \theta_{phy}} \mathcal{L}\left(\mathcal{S}, \mathbf{y}; \Phi\right), \tag{5}$$

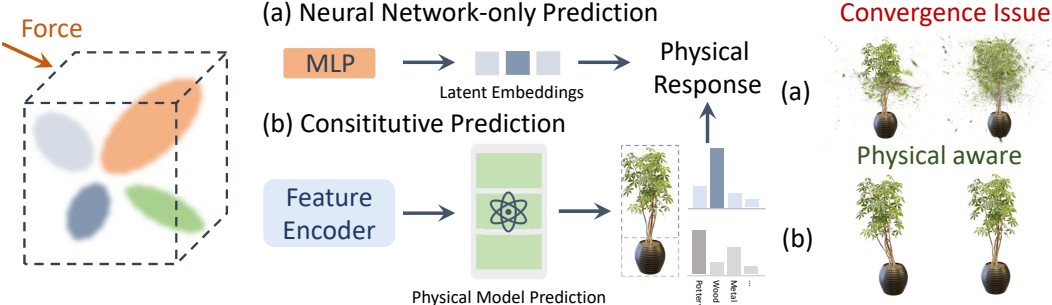

Figure 3: **Constitutive Gaussian Network.** The network architecture of Constitutive Gaussians consists of a 3D feature encoder and a physical-aware decoder. Expert-designed constitutive models are integrated into the decoder to guide the learning process, effectively avoiding the convergence issues faced by vanilla neural networks.

where $\mathcal{L}$ is the loss function to be introduced in Section 3.2.3.

Different from previous works (Zhang et al., 2024b; Liu et al., 2024; Huang et al., 2024) that keep the constitutive models fixed, Constitutive Gaussians enable the model to capture various material behaviors from both elastic and plastic deformations, thus providing a more flexible and expressive representation of the material properties. Critically, we endow a physically plausible simulating framework with the learning capability of material properties, thus achieving the pursuit of physical fidelity, general representation of diverse, heterogeneous materials, and automatic prompt-driven synthesis. The effectiveness of Constitutive Gaussians is demonstrated in experiments in Section 4.2.

### 3.2.2 PHYSICS GUIDED CONSTITUTIVE NETWORK

A naive approach to generalizing constitutive models involves training a neural network from scratch, with the expectation that the pre-trained diffusion model provides sufficient prior knowledge for the network to recover the material properties. Several previous works (Ma et al., 2023; Zong et al., 2023) have investigated replacing parts of the MPM simulator or generalizing the constitutive models with simple neural networks. However, as illustrated in Figure 3 and Section 4.3, our preliminary experiments show that neural networks without any physical priors can face difficulties in fitting the highly nonlinear constitutive models. Without ground truth for each simulated state, the NN-based MPM simulator becomes unstable and difficult to converge under the guidance of the video diffusion model. To address this issue, we propose to integrate an ensemble of expert-designed constitutive models into the neural network structure, thus utilizing more physical priors to guide the learning process. The physically guided architecture of Constitutive Gaussians consists of two main components: a 3D feature encoder backbone to extract features of 3D scenes and a physical-aware decoder to predict the material properties of Constitutive Gaussians.

**3D Feature Encoder**  We adopt a similar 3D backbone structure to PointBert (Yu et al., 2022). Our feature encoder first utilizes the Farthest Point Sampling (FPS) and k-Nearest Neighbors (kNN) algorithm to partition the particles into a set of local neighborhoods, and then applies a mini-PointNet (Qi et al., 2017a;b) to encode the information $\{\mathbf{x}_p, \mathbf{\Sigma}_p, \mathrm{sh}_p, \alpha_p\}_{p \in \mathcal{N}_i}$ of Gaussian kernels within each neighborhood $\mathcal{N}_i$ to a feature vector $\mathbf{f}_i \in \mathbb{R}^d$. The partition process is only performed once before the MPM simulation, based on the initial positions of the Gaussian kernels, thus avoiding the noise introduced by the unoptimized material properties and improving training efficiency.

**Physical-aware Decoder**  The encoded features $\mathbf{f}_i$ are then decoded by a physical-aware decoder. First, an MLP is employed to predict the material category logits for each neighborhood from a set of expert-designed constitutive models. Assuming homogeneous material properties within each neighborhood, a single expert constitutive model is assigned to all particles within each neighborhood based on the highest logit value, using a hardmax operation with a straight-through estimator (Bengio et al., 2013) as:

$$\boldsymbol{\sigma}_{p \in \mathcal{N}_i} = \Psi_{j_i}(\mathbf{F}_{p \in \mathcal{N}_i}), \ \mathbf{F}_{p \in \mathcal{N}_i} = \psi_{k_i}(\mathbf{F}_{p \in \mathcal{N}_i}^{\mathrm{trial}}), \ j_i, k_i = \arg\max\left[\mathrm{MLP}_j(\mathbf{f}_i), \mathrm{MLP}_k(\mathbf{f}_i)\right], \quad (6)$$

where the pre-defined functions $\Psi(\cdot)$ and $\psi(\cdot)$ of domain-expert models are utilized to calculate the Cauchy stress tensor $\boldsymbol{\sigma}$ and the corrected deformation gradient $\mathbf{F}$, respectively. We collect 12 combinations of constitutive models, including 3 hyperelastic density functions and 4 plasticity return functions, to cover a wide range of materials, such as pure elasticity, viscoelasticity, plasticity, and fluidity (see Appendix A.4 for details). These expert-designed constitutive models capture various material behaviors, thus providing priors that guide the neural network in learning material properties while ensuring that intermediate results adhere to physical constraints.

The intuition behind our physics-guided constitutive network is that complex scenes can be decomposed into local, homogeneous neighborhoods consisting of similar fundamental materials, which can be effectively described by expert-designed constitutive models. The hardmax operation enforces strict adherence to the physical laws, preventing the network from producing physically implausible outcomes. The physical-aware decoder is designed to be differentiable, thus enabling the end-to-end training of the entire network. Ablation studies on the network structure are provided in Section 4.3.

### 3.2.3 DIFFUSION GUIDANCE

Following previous common practice, we adopt the Score Distillation Sampling (SDS) (Poole et al., 2023) as the guidance for training Constitutive Gaussians. SDS distills 3D priors from large pre-trained text-conditioned 2D diffusion models to generate 3D content from text prompts (Lin et al., 2023; Wang et al., 2024b; Tang et al., 2023). During each training iteration, the MPM simulator starts from the initial state $s^{t_0} = \{\mathbf{x}_p^{t_0}, \mathbf{v}_p^{t_0}, \mathbf{F}_p^{t_0}, \mathbf{C}_p^{t_0}\}_{p \in \mathcal{P}}$ and steps $N-1$ times to obtain simulated states $[\hat{s}^{t_0}, \hat{s}^{t_1}, \cdots, \hat{s}^{t_{N-1}}]$. The evolutions of generated positions $\hat{\mathbf{x}}$ and deformation gradients $\hat{\mathbf{F}}$ are then passed to a MPM-compatible 3DGS renderer (see Appendix A.3) to generate a video clip $\hat{\mathcal{V}} = [\hat{I}^{t_0}, \hat{I}^{t_1}, \cdots, \hat{I}^{t_{N-1}}]$. ModelScope (Wang et al., 2023) is used as the pre-trained 2D diffusion prior $\Phi$ to supervise the video clip $\hat{\mathcal{V}}$. To clarify, noticing that each $\hat{\mathbf{x}}$ and $\hat{\mathbf{F}}$ is dependent on the model parameters $\theta = \{\theta_{el}, \theta_{pl}, \theta_{phy}\}$ when the learnable Constitutive Gaussians are integrated into the MPM simulator, the gradient of $\theta$ can be formulated as:

$$\nabla_\theta \mathcal{L}_{\text{SDS}} = \mathbb{E}_{\xi,\epsilon} \left[ \omega(\xi) \left( \hat{\epsilon}_\Phi \left( \hat{\mathcal{V}}; \xi, \mathbf{y} \right) - \epsilon \right) \frac{\partial \hat{\mathcal{V}}}{\partial \hat{\mathbf{x}}, \hat{\mathbf{F}}} \frac{\partial \hat{\mathbf{x}}, \hat{\mathbf{F}}}{\partial \theta} \right]. \tag{7}$$

### 3.2.4 TRAINING STRATEGY

We propose a novel strategy for optimizing the long simulation sequence with two key components.

**Grouping**  The MPM steps are a first-order Markov process, where the state at time $t_n$ only depends on the state at time $t_{n-1}$. To stabilize the simulating process, the time interval $\Delta t$ must be small enough, thus resulting in a large number of steps ($N \sim 1e3$) during each training iteration. This leads to high memory consumption and gradient exploding/vanishing issues given the long chain of gradient propagation. Moreover, off-the-shelf video diffusion models are not designed for such long sequences. To mitigate these issues, we first sample a subset of the simulation states uniformly and then group them into mini-batches for diffusion guidance as:

$$\hat{\mathcal{V}}_{\text{sample}} = [\underbrace{\hat{I}^{t_0}, \hat{I}^{t_0+m\Delta t}, \cdots, \hat{I}^{t_0+(M-1)m\Delta t}}_{\text{Group } 0}, \cdots, \underbrace{\cdots, \hat{I}^{t_0+(G-1)Mm\Delta t+(M-1)m\Delta t}}_{\text{Group } G-1}], \tag{8}$$

where the original video clip $\hat{\mathcal{V}}$ is sampled every $m$ frames and grouped into $G$ mini-batches with $M$ frames ($M \ll N$) in each mini-batch.

**Multiple Mini-Batch Training**  The first-order Markov property of the Material Point Method determines that each mini-batch is optimized based on the previous one. Since the video diffusion model does not directly provide the ground truth for each simulation state, the training can start from an incorrect state if the previous mini-batch is not well-optimized. Therefore, different from previous works (Liu et al., 2024; Huang et al., 2024), which optimize through the stages directly, we propose to train each mini-batch multiple times before proceeding to the next mini-batch in each training iteration. This strategy can enhance training efficiency by gradually correcting the simulation states. Ablation studies in Section 4.3 demonstrate the effectiveness of the proposed training strategy, and we provide more details of the training strategy in code snippet 1 in Appendix B.4.

Table 1: Quantitative evaluations of MPM solvers. We compare the performance of our solver with the MPM solver from NCLaw (Ma et al., 2023) on the material estimation task. We report the average time and memory consumption of a train or test iteration on $5 \times 10^4$ particles with $1 \times 10^3$ time steps. Better results are in bold.

| MPM Solver | Train Memory$_{MB}$ $\downarrow$ | Test Memory$_{MB}$ $\downarrow$ | Train Time$_s$ $\downarrow$ | Test Time$_s$ $\downarrow$ |
|---|---|---|---|---|
| NCLaw | 48,073 | **2,383** | **21.3** | 9.24 |
| Ours | **13,889** | 2,637 | 47.8 | **7.21** |

## 4 EXPERIMENTS

### 4.1 EXPERIMENTAL SETTINGS

**Simulator Details**   Off-the-shelf MPM simulators (Ma et al., 2023; Xie et al., 2024) are based on warp (Macklin, 2022), a Python library for high-performance simulation. Despite warp's compatibility with torch, the communication between these two libraries is both time-consuming and memory-intensive. To facilitate end-to-end training, we reorganized the MPM algorithm, transforming it into highly vectorized torch expressions. Table 1 shows the performance of the original MPM solvers and our implementation, demonstrating the efficiency of our method, which significantly reduces the training memory consumption, saving **75**% of GPU memory. Our implementation will be released to the public for future research.

**Datasets**   We collect several static 3DGS scenes from the public dataset of PhysGaussian (Xie et al., 2024). We also utilize BlenderNerf (Raafat, 2024) to generate more scenes with different materials and objects. Each 3D scene is generated from 100 multi-view renderings.

**Baselines**   We compare our method with three state-of-the-art 3D physical-plausible dynamic synthesis methods: (1) **PhysDreamer** (Zhang et al., 2024b), which utilizes generated video from diffusion models to supervise Young's modulus field for 3D objects; (2) **Physics3D** (Liu et al., 2024) and (3) **DreamPhysics** (Huang et al., 2024), which adopt Score Distillation Sampling to optimize Young's modulus, Poisson's ratio, and damping coefficient. Further implementation details of our method and the baselines are provided in Appendix B.1 and B.2.

**Evaluation Metrics**   Since there is no ground-truth data for dynamic 3D scenes, we propose to evaluate the performance of our method in two aspects. First, we measure the alignment between generated videos and text prompts using CLIPSIM (Wu et al., 2021). CLIPSIM is derived from the average cosine similarity between the text embedding and each frame embedding. Second, Diff$_{SSIM}$ and Diff$_{CLIP}$ are proposed to evaluate the expressiveness and robustness of our method by measuring the difference between the video generated by a trained model and a randomly initialized model with SSIM and CLIPSIM, respectively. **Higher** CLIPSIM, Diff$_{SSIM}$, and Diff$_{CLIP}$ indicate better performance. We provide calculation methods of evaluation metrics in Appendix B.3.

### 4.2 RESULTS AND COMPARISONS

**Single Object in Different Materials**   Given prompts describing different physical properties of the same object, OMNIPHYSGS can generate videos with the corresponding dynamic behaviors. For example, the prompt *a tree swinging in the wind* leads to a video where the tree is swaying back and forth, while the prompt *a tree collapsing like sand* results in a video where the tree collapses into a pile of sand. Table 2 shows the quantitative results of our method and the baselines. Our method achieves better performance in modeling all kinds of materials, surpassing current state-of-the-art methods by about **3**% ∼ **16**% across different cases in text-alignment metrics. Specifically, these baselines achieve close performance to that of our method in modeling pure elasticities, but they struggle to model the behaviors of other materials (*e.g.*, plasticity, viscoelasticity, fluid), which have different properties from those of their fixed constitutive models. This demonstrates the effectiveness and generalizability of our learnable Constitutive Gaussians in capturing the complex behaviors of different materials. Visualizations of the qualitative results are shown in Figure 4 and Appendix C.3.

Table 2: Quantitative evaluations of generating different materials for a single object. We compare our method with PhysDreamer (Zhang et al., 2024b), Physics3D (Liu et al., 2024), and Dream-Physics (Huang et al., 2024) on several cases of synthesizing different materials. **Higher** $\text{Diff}_{SSIM}$, $\text{Diff}_{CLIP}$, and CLIPSIM indicate better performance. The best results are highlighted in bold.

| Method | Scene | Swinging Ficus | Collapsing Ficus | Rubber Bear | Sand Bear | Jelly Cube | Water Cube | Average |
|---|---|---|---|---|---|---|---|---|
| PhysDreamer | $\text{Diff}_{SSIM}$ | 0.0515 | 0.0014 | 0.0620 | 0.0620 | 0.0031 | 0.0022 | 0.0303 |
| | $\text{Diff}_{CLIP}$ | 0.9771 | 1.0021 | 0.9589 | 1.0004 | 0.9590 | 1.0062 | 0.9840 |
| | $\text{CLIPSIM}_\%$ | 22.293 | 21.891 | 17.859 | 14.482 | 22.573 | 18.706 | 19.634 |
| Physics3D | $\text{Diff}_{SSIM}$ | 0.0523 | 0.0058 | 0.0640 | 0.0639 | 0.0054 | 0.0045 | 0.0327 |
| | $\text{Diff}_{CLIP}$ | 0.9574 | 0.9814 | 0.9561 | 1.0155 | 0.9658 | 1.0061 | 0.9804 |
| | $\text{CLIPSIM}_\%$ | 21.845 | 21.438 | 17.806 | 14.701 | 22.734 | 18.704 | 19.538 |
| DreamPhysics | $\text{Diff}_{SSIM}$ | 0.0516 | 0.0017 | 0.0622 | 0.0627 | 0.0038 | 0.0024 | 0.0307 |
| | $\text{Diff}_{CLIP}$ | 0.9800 | 0.9955 | 0.9522 | 0.9876 | 0.9606 | 1.0101 | 0.9810 |
| | $\text{CLIPSIM}_\%$ | 22.361 | 21.747 | 17.734 | 14.297 | 22.609 | 18.778 | 19.588 |
| Ours | $\text{Diff}_{SSIM}$ | **0.0698** | **0.1497** | **0.0763** | **0.1100** | **0.0124** | **0.0203** | **0.0731** |
| | $\text{Diff}_{CLIP}$ | **0.9929** | **1.0481** | **1.0060** | **1.1534** | **1.0006** | **1.0628** | **1.0440** |
| | $\text{CLIPSIM}_\%$ | **22.653** | **22.896** | **18.736** | **16.698** | **23.552** | **19.758** | **20.716** |

Table 3: Quantitative evaluations of generating different materials for multi-object scenes.

| Method | Scene | Rubber and Sand | Duck and Pile | Rubber hits Metal | Bear into Water | Average |
|---|---|---|---|---|---|---|
| PhysDreamer | $\text{Diff}_{SSIM}$ | 0.0737 | 0.0321 | 0.0056 | 0.0325 | 0.0360 |
| | $\text{Diff}_{CLIP}$ | 1.0397 | 1.0399 | 1.0240 | 1.0036 | 1.0268 |
| | $\text{CLIPSIM}_\%$ | 27.441 | 28.564 | 27.075 | 24.036 | 26.779 |
| Physics3D | $\text{Diff}_{SSIM}$ | 0.0566 | 0.0250 | 0.0103 | 0.0274 | 0.0298 |
| | $\text{Diff}_{CLIP}$ | 1.0333 | 1.0364 | 1.0222 | 0.9995 | 1.0228 |
| | $\text{CLIPSIM}_\%$ | 27.271 | 28.467 | 27.026 | 23.938 | 26.675 |
| DreamPhysics | $\text{Diff}_{SSIM}$ | 0.0819 | 0.0266 | 0.0261 | 0.0289 | 0.0409 |
| | $\text{Diff}_{CLIP}$ | 1.0351 | 1.0418 | 1.0436 | 0.9944 | 1.0287 |
| | $\text{CLIPSIM}_\%$ | 27.319 | 28.616 | 27.593 | 23.816 | 26.836 |
| Ours | $\text{Diff}_{SSIM}$ | **0.1129** | **0.0714** | **0.0494** | **0.0951** | **0.0822** |
| | $\text{Diff}_{CLIP}$ | **1.0570** | **1.0488** | **1.0803** | **1.0413** | **1.0569** |
| | $\text{CLIPSIM}_\%$ | **27.897** | **28.809** | **28.564** | **24.939** | **27.552** |

**Multiple Objects in Different Materials**  Facilitated by the learnable Constitutive Gaussians, OMNIPHYSGS can extract the features of each Gaussian particle within a scene and predict the material properties of each particle. This enables the model to generate dynamics with multiple objects made of different materials. Table 3 shows the quantitative results of our method and the baselines. Our method outperforms the baselines in all cases, surpassing them by about 4% in text-alignment metrics, demonstrating the ability to distinguish and model different materials for multiple objects from arbitrary prompts. Figure 5 and Appendix C.3 show the qualitative results of our method in generating dynamics with multiple objects in different materials. Notably, all baselines tend to predict homogeneous material properties for all objects in the scene, which leads to failure in modeling the dynamics of multiple objects with different materials. In contrast, our method can predict the material properties of each object in the scene according to the text prompt, highlighting the expressiveness of our Constitutive Gaussians.

**Generalization of Motion**  The Material Point Method is capable of simulating various kinds of deformation behaviors for the same scene under different boundary conditions. This endows our method with the zero-shot generalization ability to generate different motions after training on a single scene. By applying initial impulse, adding colliders, or changing the gravity direction, our method can synthesize dynamics exhibiting a wide range of material behaviors. We present qualitative results of our method's generalization ability in Appendix C.1.

Table 4: Ablation studies on network architectures (w/o physical prior) and training strategies (w/o grouping or multi-batch) of Constitutive Gaussians. We report the average $\text{Diff}_{SSIM}$, $\text{Diff}_{CLIP}$, and CLIPSIM on both the single object cases in Table 2 and the multiple object cases in Table 3.

| Scene
Method | Single Object in Different Materials | | | Multiple Objects in Different Materials | | |
|---|---|---|---|---|---|---|
| | $\text{Diff}_{SSIM}\uparrow$ | $\text{Diff}_{CLIP}\uparrow$ | $\text{CLIPSIM}_\%\uparrow$ | $\text{Diff}_{SSIM}\uparrow$ | $\text{Diff}_{CLIP}\uparrow$ | $\text{CLIPSIM}_\%\uparrow$ |
| w/o Grouping | Out of Memory | | | Out of Memory | | |
| w/o Multi-Batch | 0.0638 | 1.0224 | 20.241 | 0.0444 | 1.0067 | 26.248 |
| w/o Physical Prior | 0.0536 | 1.0241 | 20.218 | 0.0126 | 1.0025 | 23.251 |
| Ours | **0.0731** | **1.0440** | **20.716** | **0.0822** | **1.0569** | **27.552** |

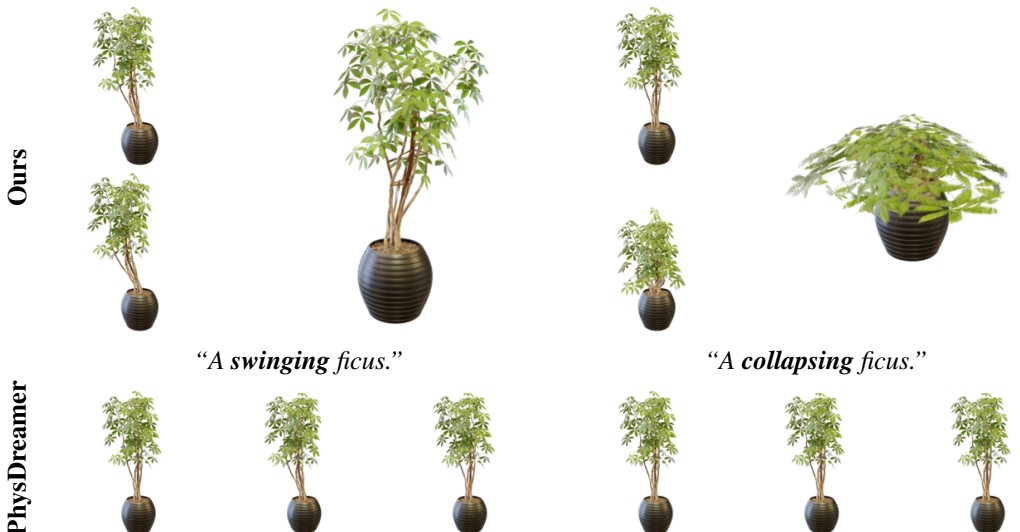

"*A swinging ficus.*"     "*A collapsing ficus.*"

Figure 4: Qualitative visualizations of 3D dynamic synthesis for a single object in different materials. We present the results of our method and PhysDreamer. Other baselines share the same problem (see Appendix C.3). The prompt is provided as the caption of each subfigure.

More qualitative results of our method and the baselines are presented in Appendix C, showing the generalization ability of our method in generating dynamics with different materials and objects. We also provide rendered videos in the supplementary material.

## 4.3 ABLATION STUDY

**Network Architecture**   We conduct ablation studies to demonstrate the necessity of introducing the expert-designed constitutive models to learnable Constitutive Gaussians. Following NCLaw (Ma et al., 2023), we replace the physical-aware decoder of Constitutive Gaussians with a simple MLP to directly predict the Cauchy stress tensor and the deformation gradient tensor (both are $3 \times 3$ matrices). Table 4 presents the quantitative results, showing that a simple MLP has difficulty fitting the highly non-linear and complex constitutive models. This underscores the importance of incorporating pre-defined physical constitutive models into the learnable Constitutive Gaussians.

**Training Strategy**   We also conduct ablation studies to investigate the effectiveness of the proposed training strategy. Specifically, we compare the performance of our method with (1) training over the whole rendered video without grouping the frames into stages and (2) training without the multi-batch strategy. Quantitative results are shown in Table 4, where training without grouping leads to *Out of Memory* due to the long chain of gradient propagation. Additionally, training without the multi-batch strategy results in a significant drop in performance due to the optimization difficulty for the first-order Markov chain. These results demonstrate that our training strategy enhances training efficiency and stability.

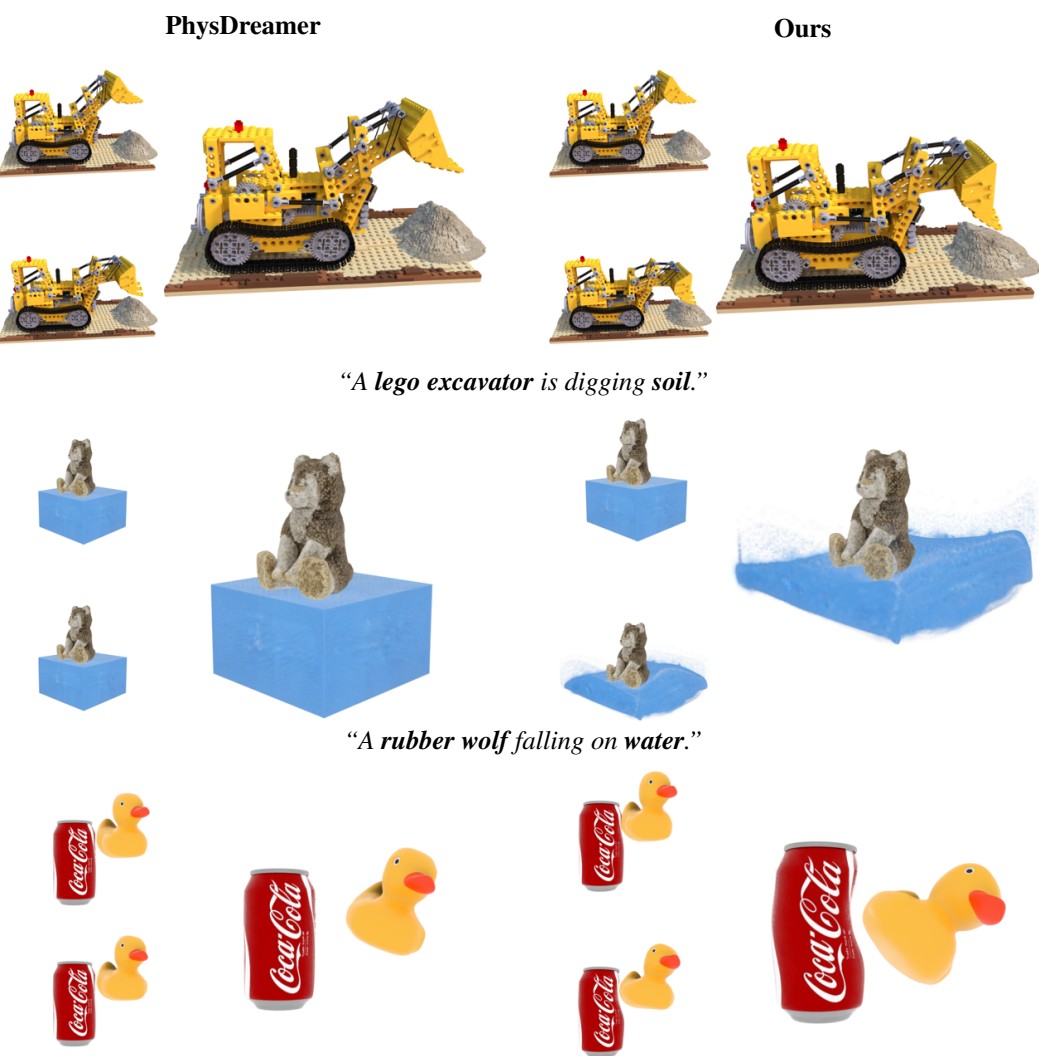

*"A **lego excavator** is digging **soil**."*

*"A **rubber wolf** falling on **water**."*

*"A **hard** duck colliding to create a dent in a **breakable metal** can."*

Figure 5: Qualitative visualizations of 3D dynamic synthesis for multiple objects in different materials. More comparison results are provided in Appendix C.3.

## 5 CONCLUSION

By introducing learnable Constitutive Gaussians, we propose a novel framework, OMNIPHYSGS, for general physics-based 3D dynamic scene synthesis. Facilitated by incorporating domain-expert constitutive models in the physics-guided network, our method can automatically and flexibly model various materials for each Gaussian particle within a scene. The supervision of pretrained text-to-video models builds a user-friendly interface for generating physically plausible and visually realistic dynamic scenes aligned with text prompts. We hope our work could be beneficial in practical scenarios, such as immersive video games, hyper-realistic virtual reality experiences, robotic simulations, and computer-aided design tools.

**Limitations and Future Work** In this work, we pick expert-designed constitutive models limited to several representative materials, and the per-scene SDS optimization is time-intensive. Future work could consider collecting more kinds of materials to scale up the diversity of the materials and designing a more efficient optimization algorithm to build an amortized model for instant inference.

## REPRODUCIBILITY STATEMENT

To enhance the reproducibility of our work, we provide detailed experimental settings, including dataset sources in Section 4.1, baseline implementations in Appendix B.2, and evaluation metrics in Appendix B.3. Training details, such as the hyperparameters, are outlined in Appendix B.1, and the code snippets for the training process are provided in Appendix B.4. A detailed description of the Material Point Method algorithm is available in Appendix A. Our code and data will be released after curation.

## ACKNOWLEDGMENTS

The work was supported by National Key R&D Program of China (2022ZD0160300), an internal grant of Peking University (2024JK28), and a grant from Kuaishou (No. DJHL-20240809-115).

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

# APPENDIX

## A  MATERIAL POINT METHOD

### A.1  CONTINUUM MECHANICS

Continuum mechanics (Gonzalez & Stuart, 2008; Jiang et al., 2016) models the behavior of materials as a continuous medium by a deformation map $\mathbf{x} = \phi(\mathbf{X}, t)$ where $\mathbf{X}$ denotes the undeformed configuration and $\mathbf{x}$ denotes the deformed configuration. The deformation gradient $\mathbf{F} = \nabla_{\mathbf{X}} \phi(\mathbf{X}, t)$ describes the local deformation of the material. The key governing equations for the deformation of a material are the conservation of mass and momentum:

$$\frac{\mathrm{D}\rho}{\mathrm{D}t} + \rho \nabla \cdot \mathbf{v} = 0, \quad \rho \frac{\mathrm{D}\mathbf{v}}{\mathrm{D}t} = \nabla \cdot \boldsymbol{\sigma} + \mathbf{f}^{\text{ext}} \tag{9}$$

where $\rho(\mathbf{x}, t)$ is the density, $\mathbf{v}(\mathbf{x}, t)$ is the velocity field, $\boldsymbol{\sigma}(\mathbf{x}, t)$ is the Cauchy stress tensor, and $\mathbf{f}^{\text{ext}}(\mathbf{x}, t)$ is the external force. The Cauchy stress tensor is defined as

$$\boldsymbol{\sigma}(\mathbf{x}, t) = \frac{1}{\det(\mathbf{F})} \frac{\partial \Psi}{\partial \mathbf{F}} (\mathbf{F}^E) \mathbf{F}^{E^T} \in \mathbb{R}^{3 \times 3}, \tag{10}$$

where $\Psi : \mathbb{R}^{3 \times 3} \to \mathbb{R}^{3 \times 3}$ is the hyperelastic energy density function, $\mathbf{F}^E \in \mathbb{R}^{3 \times 3}$ is the elastic part of the deformation gradient, since the total deformation gradient can be decomposed as $\mathbf{F} = \mathbf{F}^E \mathbf{F}^P$ where $\mathbf{F}^P \in \mathbb{R}^{3 \times 3}$ is the plastic part of the deformation gradient. In practice, the decomposition equation of the deformation gradient can be reparameterized as $\mathbf{F} = \psi(\mathbf{F}^E)$ where $\psi : \mathbb{R}^{3 \times 3} \to \mathbb{R}^{3 \times 3}$ is a return function that applies the plasticity constraints to $\mathbf{F}^E$.

The hyperelastic energy density function $\Psi(\cdot)$ and the plasticity return function $\psi(\cdot)$ are the constitutive models that describe the material behavior. They map the elastic part of the deformation gradient $\mathbf{F}^E$ to the Cauchy stress tensor $\boldsymbol{\sigma}$ and the corrected deformation gradient $\mathbf{F} \in \mathbb{R}^{3 \times 3}$, respectively. These models are typically designed by experts in the field to describe the material properties, determining how the material deforms under certain circumstances. To match the material behavior, these functions are highly nonlinear and can be of great complexity. Appendix A.4 provides some examples of these models, showing that constitutive models can be applied to a wide range of materials.

### A.2  MPM ALGORITHM

MPM (Jiang et al., 2016; Stomakhin et al., 2013) is a hybrid Eulerian-Lagrangian method that discretizes the continuum into a set of particles and utilizes a background grid to solve the conservation equations. As shown in the pseudo-code in Algorithm 1, given boundary conditions $\mathbf{b}$ (*e.g.* initial velocity, impulse, external force, etc.), MPM performs a particle-to-grid (P2G) step to transfer the particle information (*i.e.* the mass and momentum) to the grid and a grid-to-particle (G2P) step to transfer the grid information back to the particles in the simulation loop.

---

**Algorithm 1** A Moving Least Squares Material Point Method (MLS-MPM) Step

---

**Input:** $s^{t_n} = \{\mathbf{x}_p^{t_n}, \mathbf{v}_p^{t_n}, \mathbf{F}_p^{t_n}, \mathbf{C}_p^{t_n}\}_{p \in \mathcal{P}}$
**Output:** $s^{t_{n+1}} = \{\mathbf{x}_p^{t_{n+1}}, \mathbf{v}_p^{t_{n+1}}, \mathbf{F}_p^{t_{n+1}}, \mathbf{C}_p^{t_{n+1}}\}_{p \in \mathcal{P}}$
 1: **for all** $p \in \mathcal{P}$ **do**
 2:     $\text{stress}_p^{t_{n+1}} \leftarrow \texttt{F2Stress}(\mathbf{F}_p^{t_n}, \Psi_p, \boldsymbol{\gamma} a_p)$
 3:     $\mathbf{x}_p^{t_n}, \mathbf{v}_p^{t_n} \leftarrow \texttt{ApplyBoundaryConditions}(\mathbf{x}_p^{t_n}, \mathbf{v}_p^{t_n}, \mathbf{b})$
 4:     $\mathbf{x}_p^{t_{n+1}}, \mathbf{v}_p^{t_{n+1}}, \mathbf{F}_{p, \text{ trial}}^{t_{n+1}}, \mathbf{C}_p^{t_{n+1}} \leftarrow \texttt{Particle2Grid2Particle}(\mathbf{x}_p^{t_n}, \mathbf{v}_p^{t_n}, \mathbf{C}_p^{t_n}, \text{stress}_p^{t_{n+1}})$
 5:     $\mathbf{F}_p^{t_{n+1}} \leftarrow \texttt{ReturnMap}(\mathbf{F}_{p, \text{ trial}}^{t_{n+1}}, \psi_p, \boldsymbol{\gamma}_p)$
 6: **end for**

---

MPM traces particles' states $s^{t_n} = \{\mathbf{x}_p^{t_n}, \mathbf{v}_p^{t_n}, \mathbf{F}_p^{t_n}, \mathbf{C}_p^{t_n}\}_{p \in \mathcal{P}}$ in every time step $t_n$, where $\mathbf{x}_p^{t_n} \in \mathbb{R}^3$ is the position of the particle, $\mathbf{v}_p^{t_n} \in \mathbb{R}^3$ is the velocity, $\mathbf{F}_p^{t_n} \in \mathbb{R}^{3 \times 3}$ is the deformation gradient, and $\mathbf{C}_p^{t_n} \in \mathbb{R}^{3 \times 3}$ is the affine momentum. The hyperelastic energy density function $\Psi_p(\cdot)$ and the

plasticity return function $\psi_p(\cdot)$ are utilized as the **constitutive models** to calculate the stress tensor and to correct the trial deformation gradient, respectively. Physical parameters $\boldsymbol{\gamma}_p \in \mathbb{R}^K$, such as Young's modulus and Poisson's ratio, are included in the constitutive models.

We denote $m_p \in \mathbb{R}$ and $V_p \in \mathbb{R}$ as the mass of particle $p$ and the volume of the particle, respectively, which are invariant during the dynamics. For each grid node $i \in \mathcal{G}$, we denote $m_i^{t_n} \in \mathbb{R}$ and $\mathbf{v}_i^{t_n} \in \mathbb{R}^3$ as the mass and velocity of the node at time $t_n$, respectively. The position of each grid node is denoted as $\mathbf{x}_i \in \mathbb{R}^3$, which is fixed during the simulation. Mass and momentum are transferred between particles and grid nodes, according to the interpolation functions $\mathcal{N}_i(\mathbf{x})$. We denote the interpolation result of particle $p$ at grid node $i$ as $\omega_{ip}^{t_n} \in \mathbb{R}$.

In practice, we prefer to express the strain-stress relationship using deformation gradient $\mathbf{F}_p$ and first Piola-Kirchoff stress tensor $\mathbf{P}_p \in \mathbb{R}^{3 \times 3}$ of a particle $p$ as:

$$\mathbf{P}_p = \frac{\partial \Psi_p}{\partial \mathbf{F}_p}, \tag{11}$$

because they are more naturally expressed in the material space (Jiang et al., 2016). For simplicity, we **omit the partial derivative symbol** in the following equations. The MPM algorithm is summarized as follows (items 1∼5 illustrate a single MPM time step):

0. **Initialization** Initialize the particle state from the static Gaussians as:

$$\mathbf{x}_p^{t_0} = \mathbf{x}_p, \quad \mathbf{v}_p^{t_0} = \mathbf{0}, \quad \mathbf{F}_p^{t_0} = \mathbf{I}, \quad \mathbf{C}_p^{t_0} = \mathbf{0}, \quad \forall p \in \mathcal{P}, \tag{12}$$

and set the grid mass and velocity to zero as:

$$m_i^{t_0} = 0, \quad \mathbf{v}_i^{t_0} = \mathbf{0}, \quad \forall i \in \mathcal{G} \tag{13}$$

1. **Stress Update** Calculate the stress tensor $\mathbf{P}_p^{t_n}$ as:

$$\mathbf{P}_p^{t_n} = \Psi_p\left(\mathbf{F}_p^{t_n}\right), \quad \forall p \in \mathcal{P} \tag{14}$$

2. **Particle to Grid (P2G)** Transfer particle information to grids using APIC scheme (Jiang et al., 2015) as:

$$m_i^{t_n} = \sum_{p \in \mathcal{P}} \omega_{ip}^{t_n} m_p, \quad \forall i \in \mathcal{G} \tag{15}$$

$$m_i^{t_n} \mathbf{v}_i^{t_n} = \sum_{p \in \mathcal{P}} \omega_{ip}^{t_n} m_p \left(\mathbf{v}_p^{t_n} + \mathbf{C}_p^{t_n}\left(\mathbf{x}_i - \mathbf{x}_p^{t_n}\right)\right) \tag{16}$$

3. **Grid Update** Apply internal and external forces to the grid nodes as:

$$\mathbf{v}_i^{t_{n+1}} = \mathbf{v}_i^{t_n} - \frac{\Delta t}{m_i} \sum_{p \in \mathcal{P}} \mathbf{P}_p \nabla \omega_{ip}^{t_n} V_p + \Delta t \mathbf{f}_{\text{ext}}, \quad \forall i \in \mathcal{G} \tag{17}$$

4. **Grid to Particle (G2P)** Transfer grid information to particles as:

$$\mathbf{v}_p^{t_{n+1}} = \sum_{i \in \mathcal{G}} \omega_{ip}^{t_n} \mathbf{v}_i^{t_{n+1}}, \quad \mathbf{x}_p^{t_{n+1}} = \mathbf{x}_p^{t_n} + \Delta t \mathbf{v}_p^{t_{n+1}}, \tag{18}$$

$$\mathbf{C}_p^{t_{n+1}} = \frac{12}{\Delta x^2 (b+1)} \sum_{i \in \mathcal{G}} \omega_{ip}^{t_n} \mathbf{v}_i^{t_{n+1}} \left(\mathbf{x}_i - \mathbf{x}_p^{t_n}\right)^T, \quad \forall p \in \mathcal{P} \tag{19}$$

$$\nabla \mathbf{v}_p^{t_{n+1}} = \sum_{i \in \mathcal{G}} \mathbf{v}_i^{t_{n+1}} \nabla \omega_{ip}^{t_n T}, \quad \mathbf{F}_{p, \text{trial}}^{t_{n+1}} = \left(\mathbf{I} + \Delta t \nabla \mathbf{v}_p^{t_{n+1}}\right) \mathbf{F}_p^{t_n}, \tag{20}$$

where $b$ is the B-spline degree and $\Delta x$ is the grid spacing.

5. **Plasticity Correction** Apply the plasticity return function to correct the trial deformation gradient as:

$$\mathbf{F}_p^{t_{n+1}} = \psi_p\left(\mathbf{F}_{p, \text{trial}}^{t_{n+1}}\right), \quad \forall p \in \mathcal{P} \tag{21}$$

Boundary conditions can be applied to particles before P2G and to grid nodes before G2P on the positions or velocities. Readers can refer to (Jiang et al., 2016; Stomakhin et al., 2013) for more details about the MPM algorithm.

### A.3 UPDATES FOR RENDERING PROCESS

Due to the point-wise nature of the MPM, it is well-suited for simulating 3D Gaussian particles given the evolutions of the Gaussian kernels' positions and deformation gradients. Following Phys-Gaussian (Xie et al., 2024), we update Gaussian kernels at each time step to ensure that the kernels follow the Gaussian distribution after deformation as:

$$\mathbf{x}_{p,render}^{t_n} \leftarrow \mathbf{x}_p^{t_n}, \quad \boldsymbol{\Sigma}_{p,render}^{t_n} \leftarrow \mathbf{F}_p^{t_n} \boldsymbol{\Sigma}_p \left(\mathbf{F}_p^{t_n}\right)^T. \tag{22}$$

The rendering view direction $\mathbf{d}_p$ from the viewpoint to each Gaussian kernel is also modified by the deformation gradient as:

$$\mathbf{d}_{p,render}^{t_n} \leftarrow \left(\mathbf{R}_p^{t_n}\right)^T \mathbf{d}_p, \tag{23}$$

where $\mathbf{R}_p$ is derived from the polar decomposition of the deformation gradient $\mathbf{F}_p = \mathbf{R}_p \mathbf{S}_p$. Other necessary attributes of the Gaussian kernels, such as spherical harmonic coefficients and opacity, are kept unchanged during the simulation process. We render the Gaussian kernels at each time step using the official implementation of Gaussian Splatting renderer (Kerbl et al., 2023)[1].

### A.4 CONSTITUTIVE MODELS

We collect expert-designed constitutive models from previous works (Ma et al., 2023; Zong et al., 2023; Xie et al., 2024), which describe several representative materials including materials with elasticity (*e.g.*, rubber, branches, and cloth), plasticity (*e.g.*, snow, metal, and clay), viscoelasticity (*e.g.*, honey and mud), and fluidity (*e.g.*, water, oil, and lava). Table 5 lists some necessary physical parameters used in the constitutive models.

Table 5: Physical Parameters for Constitutive Models.

| Notation | $E$ | $\nu$ | $\mu$ | $\lambda$ |
|---|---|---|---|---|
| Description | Young's Modulus | Poisson's Ratio | Shear Modulus | Lamé Parameter |
| Value | $E$ | $\nu$ | $\frac{E}{2(1+\nu)}$ | $\frac{E\nu}{(1+\nu)(1-2\nu)}$ |

#### A.4.1 HYPERELASTIC ENERGY DENSITY FUNCTION

We use the first Piola-Kirchoff stress tensor $\mathbf{P} = \frac{\partial \Psi}{\partial \mathbf{F}}$ to express the stress-strain relationship by listing the map from deformation gradient $\mathbf{F}$ to $\mathbf{P}$.

**Fixed Corotated Elasticity**    We follow (Stomakhin et al., 2012) to define the fixed corotated elasticity as:

$$\mathbf{P}(\mathbf{F}) = 2\mu(\mathbf{F} - \mathbf{R}) + \lambda J (J - 1) \mathbf{F}^{-T}, \tag{24}$$

where $\mathbf{R}$ is from the polar decomposition of $\mathbf{F} = \mathbf{R}\mathbf{S}$ and $J$ is the determinant of $\mathbf{F}$. Fixed corotated elasticity is suitable for simulating rubber-like materials.

**Neo-Hookean Elasticity**    We follow (Bonet & Wood, 1997) to define the Neo-Hookean elasticity as:

$$\mathbf{P}(\mathbf{F}) = \mu\left(\mathbf{F} - \mathbf{F}^{-T}\right) + \lambda \log(J) \mathbf{F}^{-T}. \tag{25}$$

Neo-Hookean elasticity is suitable for simulating elasticities like springs.

**StVK Elasticity**    We follow (Klár et al., 2016; Barbič & James, 2005) to define the StVK elasticity as:

$$\mathbf{P}(\mathbf{F}) = \mathbf{U}\left(2\mu\boldsymbol{\Sigma}^{-1}\ln\boldsymbol{\Sigma} + \lambda\text{tr}(\ln\boldsymbol{\Sigma})\boldsymbol{\Sigma}^{-1}\right)\mathbf{V}^T, \tag{26}$$

where $\mathbf{U}$, $\boldsymbol{\Sigma}$, and $\mathbf{V}$ are derived from the singular value decomposition of $\mathbf{F} = \mathbf{U}\boldsymbol{\Sigma}\mathbf{V}^T$. StVK elasticity is suitable for simulating plasticity like sand and metal.

---

[1] https://github.com/graphdeco-inria/gaussian-splatting

### A.4.2 PLASTICITY RETURN FUNCTION

We use the plasticity return function $\psi\left(\cdot\right)$ to correct the trial deformation gradient $\mathbf{F}_{\text{trial}}$ to the final deformation gradient $\mathbf{F}$. For simplicity, we **omit the subscript** $trial$ in the following equations.

**Identity Plasticity**    Most pure elastic materials adopt the identity plasticity as:

$$\psi\left(\mathbf{F}\right) = \mathbf{F}. \tag{27}$$

**Drucker-Prager Plasticity**    We follow (Drucker & Prager, 1952; Klár et al., 2016; Yue et al., 2018; Chen et al., 2021) to define the Drucker-Prager plasticity as:

$$\psi\left(\mathbf{F}\right) = \mathbf{U}\mathcal{Z}\left(\mathbf{\Sigma}\right)\mathbf{V}^{T}, \quad \mathcal{Z}\left(\mathbf{\Sigma}\right) = \begin{cases} \mathbf{I}, & \text{sum}\left(\boldsymbol{\epsilon}\right) > 0, \\ \mathbf{\Sigma}, & \delta\boldsymbol{\gamma} \leq 0 \text{ and sum}\left(\boldsymbol{\epsilon}\right) \leq 0, \\ \exp\left(\boldsymbol{\epsilon} - \delta\boldsymbol{\gamma}\frac{\hat{\boldsymbol{\epsilon}}}{\|\hat{\boldsymbol{\epsilon}}\|}\right), & \text{otherwise}, \end{cases} \tag{28}$$

where $\boldsymbol{\epsilon} = \log\left(\mathbf{\Sigma}\right)$. Drucker-Prager plasticity is suitable for simulating plasticity like snow and sand.

**von Mises Plasticity**    We follow (Mises, 1913; Hu et al., 2018; Huang et al., 2021) to define the von Mises plasticity as:

$$\psi\left(\mathbf{F}\right) = \mathbf{U}\mathcal{Z}\left(\mathbf{\Sigma}\right)\mathbf{V}^{T}, \quad \mathcal{Z}\left(\mathbf{\Sigma}\right) = \begin{cases} \mathbf{\Sigma}, & \delta\boldsymbol{\gamma} \leq 0, \\ \exp\left(\boldsymbol{\epsilon} - \delta\boldsymbol{\gamma}\frac{\hat{\boldsymbol{\epsilon}}}{\|\hat{\boldsymbol{\epsilon}}\|}\right), & \text{otherwise}. \end{cases} \tag{29}$$

von Mises plasticity is suitable for simulating plasticity like metal and clay.

**Fluid Plasticity**    We follow (Stomakhin et al., 2014; Gao et al., 2018) to define the fluid plasticity as:

$$\psi\left(\mathbf{F}\right) = J^{1/3}\mathbf{I}. \tag{30}$$

Fluid plasticity is suitable for simulating fluidity like water and lava.

## B MORE DETAILS ON IMPLEMENTATION

### B.1 HYPERPARAMETERS AND TRAINING SETTINGS

**Simulating Details**    We set the simulating timestep $\Delta t$ to $3\times10^{-4}$ for all experiments. In training, the simulated states are sampled every 10 steps and rendered into images with a fixed camera. The video length is set to 150 frames with a frame rate of 30 fps, resulting in a total of 5 seconds. We load all Gaussian particles in a $1 \times 1 \times 1$ bounding box, which is divided into $25 \times 25 \times 25$ grids. The simulating world is equipped with a gravity of $9.8$ m/s$^2$. In the training phase, we do not apply any other initial velocity or external force to the particles for all quantitative experiments except the *swinging ficus* case.

**Training Details**    All experiments are conducted on a single NVIDIA A6000 (48GB) GPU. We train our model using the Adam optimizer (Kingma, 2014) with a learning rate of $5 \times 10^{-5}$ for 5 iterations of each scene. In each iteration, the whole simulating process is divided into 10 stages sequentially where 15 frames are rendered for each stage. Video clips of each stage are optimized by a pretrained text-to-video diffusion model (Wang et al., 2023) for 30 iterations before we proceed to the next stage. For the learnable Constitutive Gaussians, we first adopt FPS to sample 8192 centers from the original Gaussian particles and then use a convolutional layer to encode the features within a neighborhood and a simple MLP to predict the category of the material. Each neighborhood includes 32 particles.

### B.2 BASELINE DETAILS

We reimplement the baseline methods according to their GitHub repositories.

- **PhysDreamer** (Zhang et al., 2024b)[2], which utilizes diffusion-generated reference videos to optimize Young's modulus in constitutive combinations of Fixed Corotated Elasticity and Identity Plasticity. We use our text prompts to generate the reference videos as training data for PhysDreamer.

- **Physics3D** (Liu et al., 2024)[3], which adopts StVK Elasticity and a dissipation term as constitutive models. SDS is used to optimize Young's modulus, Poisson's ratio, and damping coefficient in Physics3D.

- **DreamPhysics** (Huang et al., 2024)[4], which leverages SDS to optimize Young's modulus and Poisson's ratio in Fixed Corotated Elasticity and Identity Plasticity.

We use the same experimental settings across all methods. Since our multi-batch training strategy actually trains the model multiple times in a single iteration, we train the baselines for `number of iterations` × `number of internal iterations` iterations to ensure a fair comparison.

### B.3 EVALUATION DETAILS

We use the `Vit-L/14` version of CLIP (Radford et al., 2021) to calculate the CLIPSIM (Wu et al., 2021) score as:

$$\text{CLIPSIM} = \frac{1}{N} \sum_{n=1}^{N} CLIP(\hat{I}_n, \mathbf{y}) \tag{31}$$

where $\hat{I}_n$ is the $n$-th frame of the generated video and $\mathbf{y}$ is the text prompt. Higher CLIPSIM indicates better alignment between the video and the text. $\text{Diff}_{SSIM}$ and $\text{Diff}_{CLIP}$ are derived from:

$$\text{Diff}_{SSIM} = 1 - \frac{1}{N} \sum_{n=1}^{N} \text{SSIM}(I'_n, \hat{I}_n), \quad \text{Diff}_{CLIP} = \frac{\text{CLIPSIM}}{\text{CLIPSIM}'} \tag{32}$$

where $I'_n$ is the $n$-th frame of the video generated by a randomly initialized model, $SSIM$ is the structural similarity index (Wang et al., 2004), and $\text{CLIPSIM}'$ is the CLIPSIM of the random model. These two metrics, $\text{Diff}_{SSIM}$ and $\text{Diff}_{CLIP}$, measure the improvement of the model during training and eliminate the influence of initialization. Higher $\text{Diff}_{SSIM}$ and $\text{Diff}_{CLIP}$ indicate better learning ability and robustness.

### B.4 TRAINING STRATEGY

Code snippet 1 illustrates our training strategy of first dividing the whole simulating process into multiple stages, grouping the frames of each stage into mini-batches, and then optimizing the video clips of each mini-batch multiple times. Note that each frame is sampled every fixed number of steps.

## C ADDITIONAL RESULTS

### C.1 VISUALIZATIONS OF MOTION GENERALIZATION

Given different boundary conditions, our trained model can be generalized to synthesize diverse dynamic behaviors. We present additional visualizations of motion generalization in Figure 6, taking the *rubber wolf* as an example.

### C.2 VISUALIZATIONS OF OMNIPHYSGS

We present additional visualizations of 3D dynamic synthesis for a single object in different materials in Figure 7 and for multiple objects in Figure 8.

---

[2]https://github.com/a1600012888/PhysDreamer
[3]https://github.com/liuff19/Physics3D
[4]https://github.com/tyhuang0428/DreamPhysics

Listing 1: Training Strategy

```python
# Train an iteration
x, v, F, C = initialize()
for stage in range(num_stages):
    # Grouping
    # Save the end state of the last stage for multi-batch
    x_ckpt, v_ckpt, F_ckpt, C_ckpt = x, v, F, C
    for internal_iteration in range(num_internal_iterations):
        # Multi-Batch
        # Start from the end state of the last stage
        x, v, F, C = x_ckpt, v_ckpt, F_ckpt, C_ckpt
        stage_frames = []
        for frame in range(num_frames):
            for step in range(num_steps_per_frame):
                # MPM steps
                x, v, F, C = mpm_step(x, v, F, C)
            rendering = render(x, v, F, C)
            stage_frames.append(rendering)
        loss = score_distillation_sampling(stage_frames)
        loss.backward()
        optimizer.step()
```

## C.3 VISUAL COMPARISONS WITH BASELINES

We present visual comparisons with baselines of 3D dynamic synthesis for a single object in different materials in Figures 9, 10, and 11, and for multiple objects in Figures 12 and 13. Remarkably, all baselines tend to predict the same material regardless of the different input prompts, resulting in dynamic synthesis outputs that appear quite similar. This highlights that merely tuning physical parameters is insufficient for synthesizing diverse dynamic behaviors, constrained by the limited expressiveness of the fixed physics model. In contrast, our method effectively synthesizes a wide range of dynamic behaviors by learning the material properties from pretrained video diffusion, thus achieving better generalizability and expressiveness.

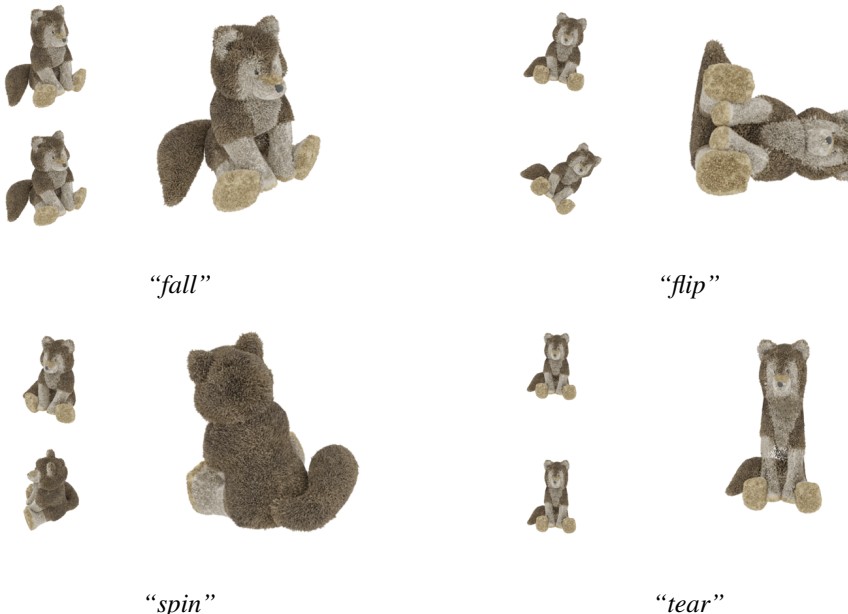

*"fall"*                                    *"flip"*

*"spin"*                                    *"tear"*

Figure 6: Visualization results of motion generalization.

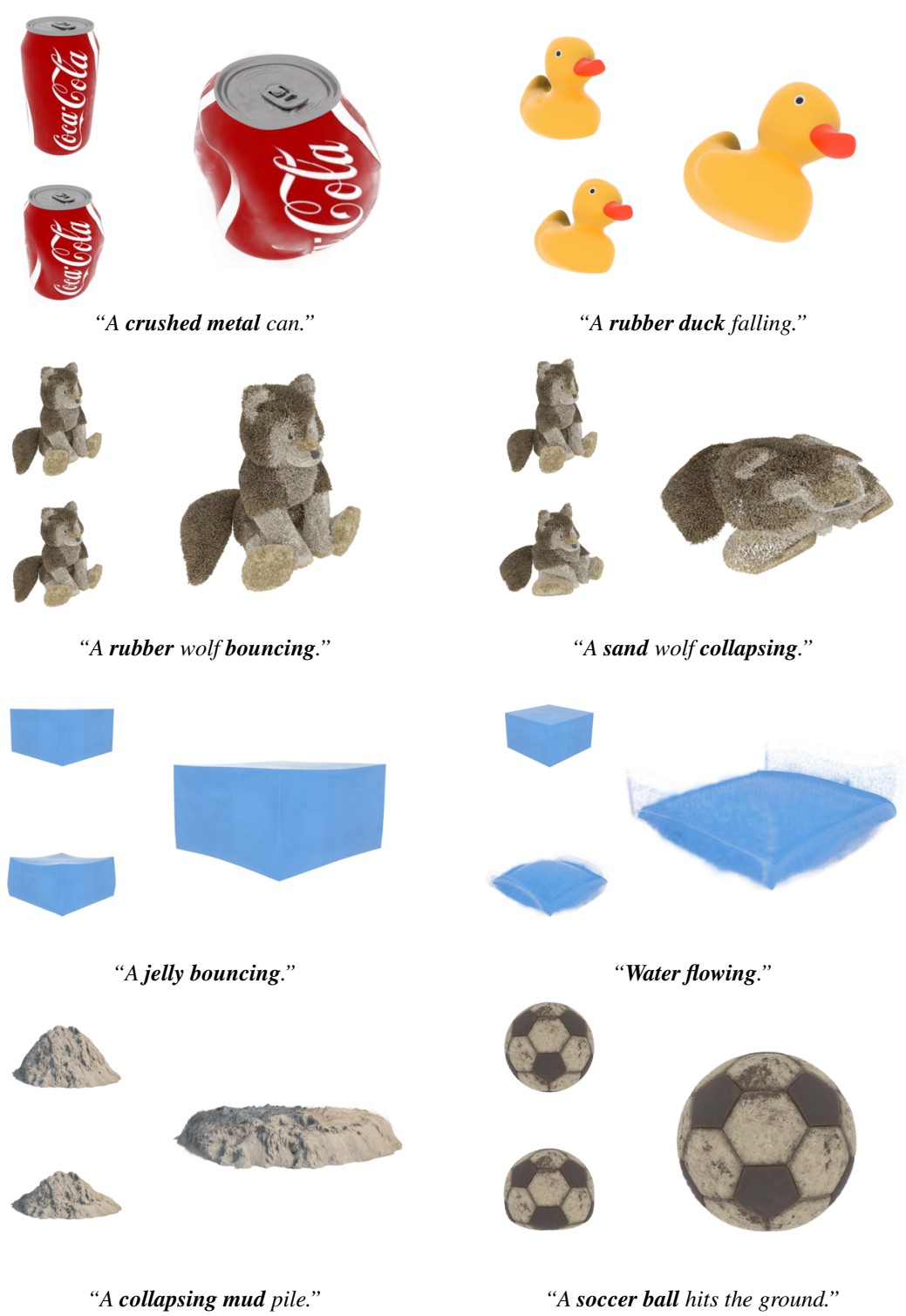

Figure 7: Qualitative visualizations of 3D dynamic synthesis for a single object with our method.

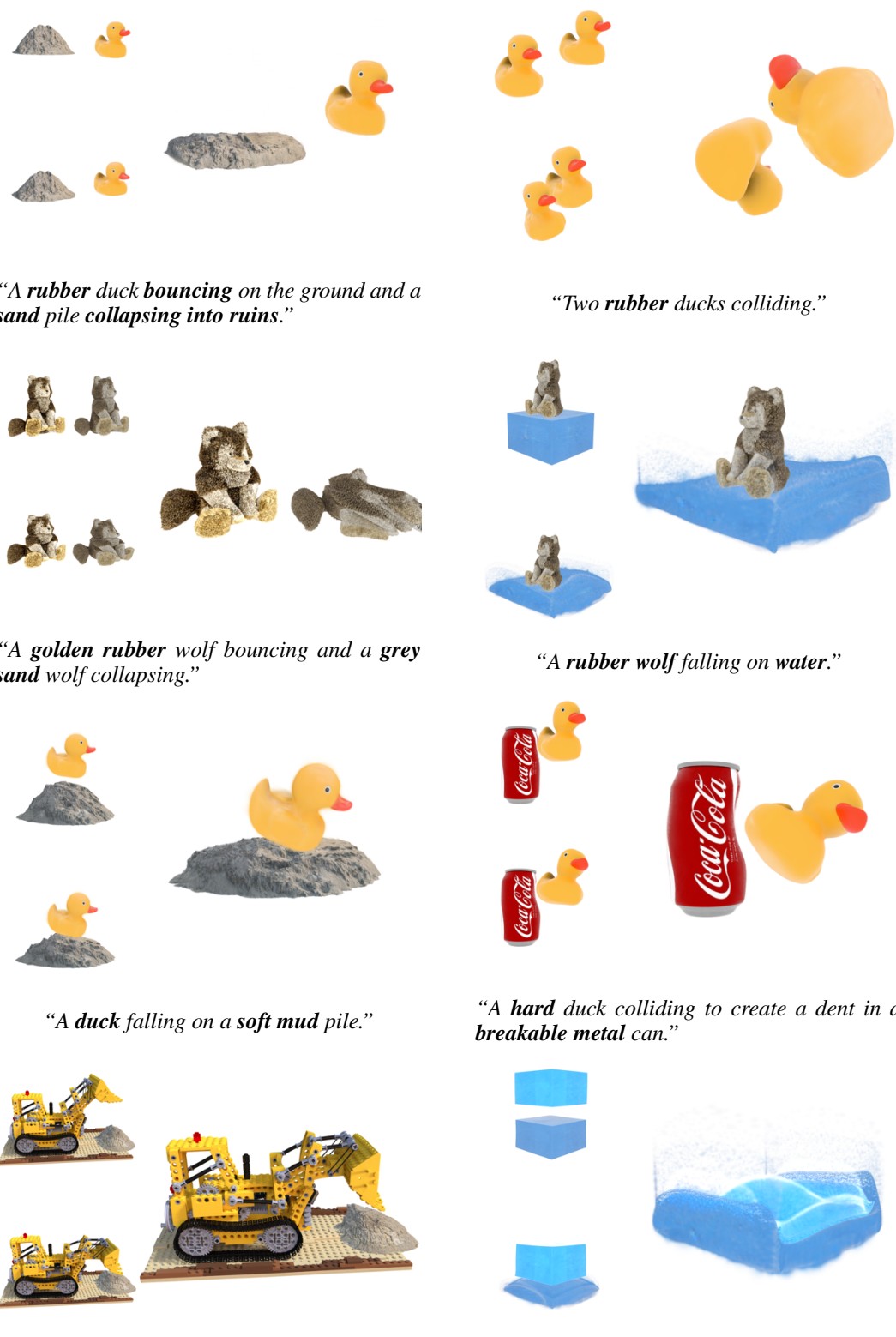

*"A **rubber** duck **bouncing** on the ground and a **sand** pile **collapsing into ruins**."*

*"Two **rubber** ducks colliding."*

*"A **golden rubber** wolf bouncing and a **grey sand** wolf collapsing."*

*"A **rubber wolf** falling on **water**."*

*"A **duck** falling on a **soft mud** pile."*

*"A **hard** duck colliding to create a dent in a **breakable metal** can."*

*"A **lego excavator** is digging **soil**."*

*"Two kinds of **water** flowing."*

Figure 8: Qualitative visualizations of 3D dynamic synthesis for multiple objects with our method.

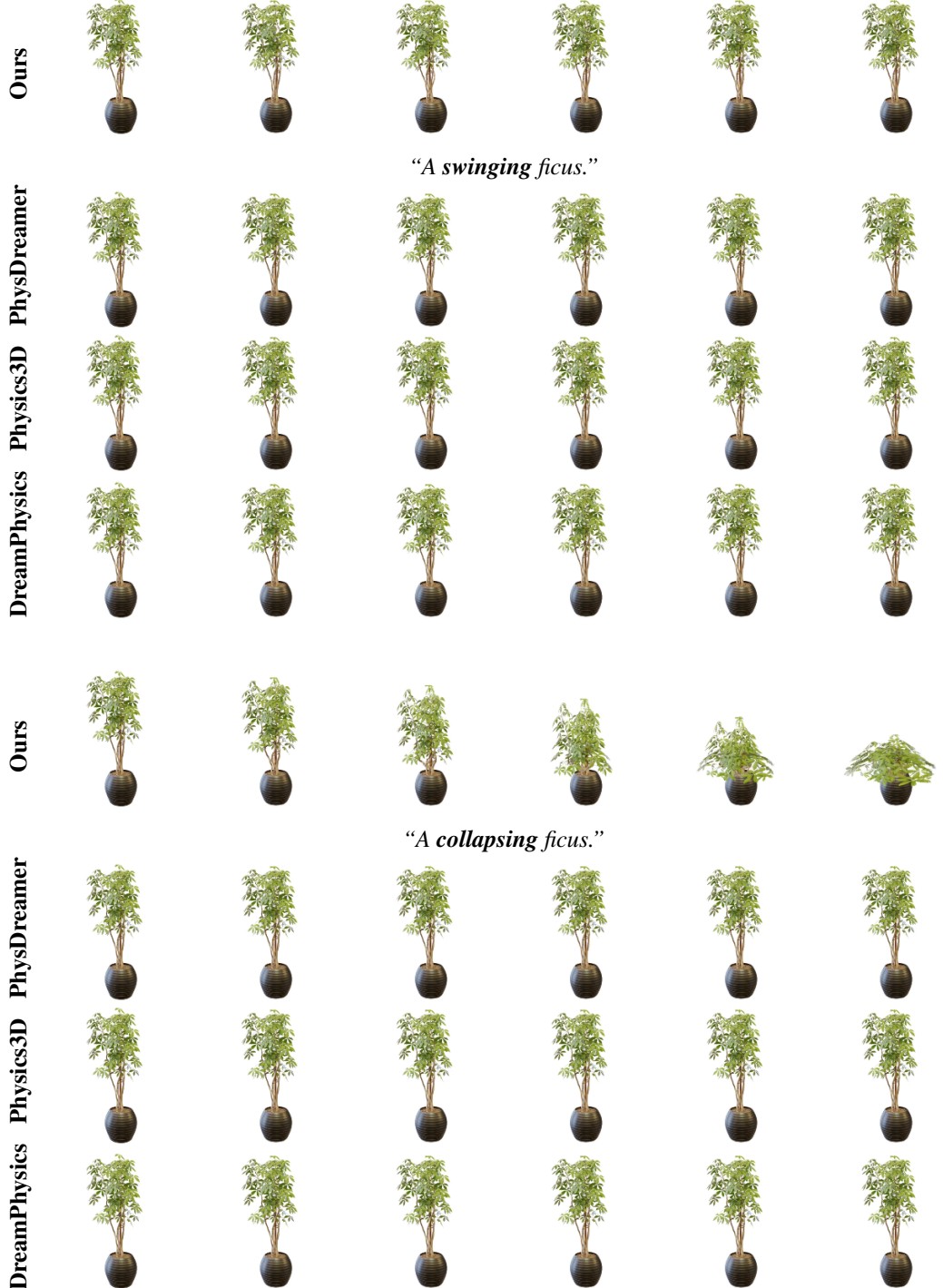

Figure 9: Qualitative visualizations of 3D dynamic synthesis for a single object in different materials. We present the results of our method and the baselines.

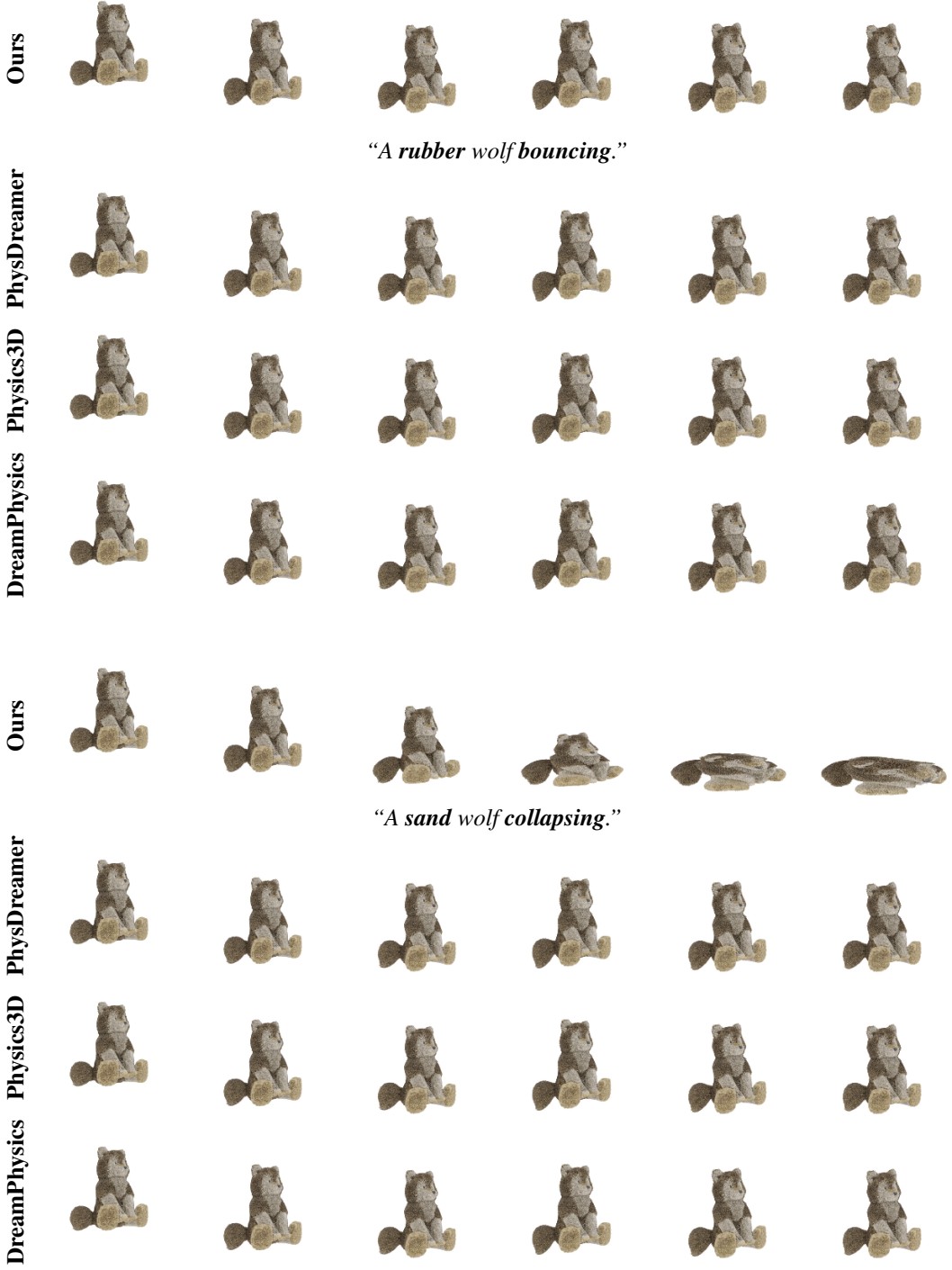

Figure 10: Qualitative visualizations of 3D dynamic synthesis for a single object in different materials. We present the results of our method and the baselines.

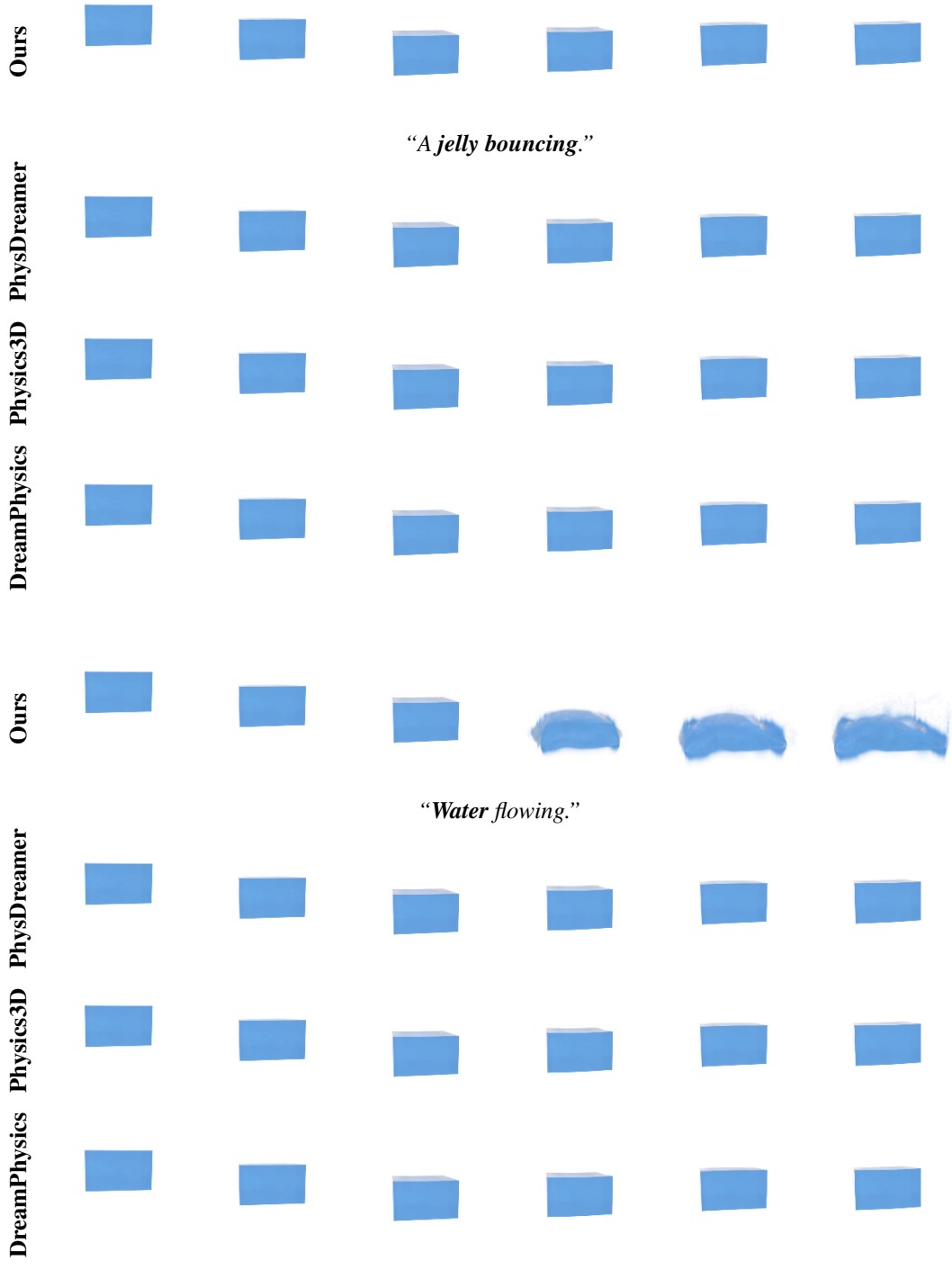

Figure 11: Qualitative visualizations of 3D dynamic synthesis for a single object in different materials. We present the results of our method and the baselines.

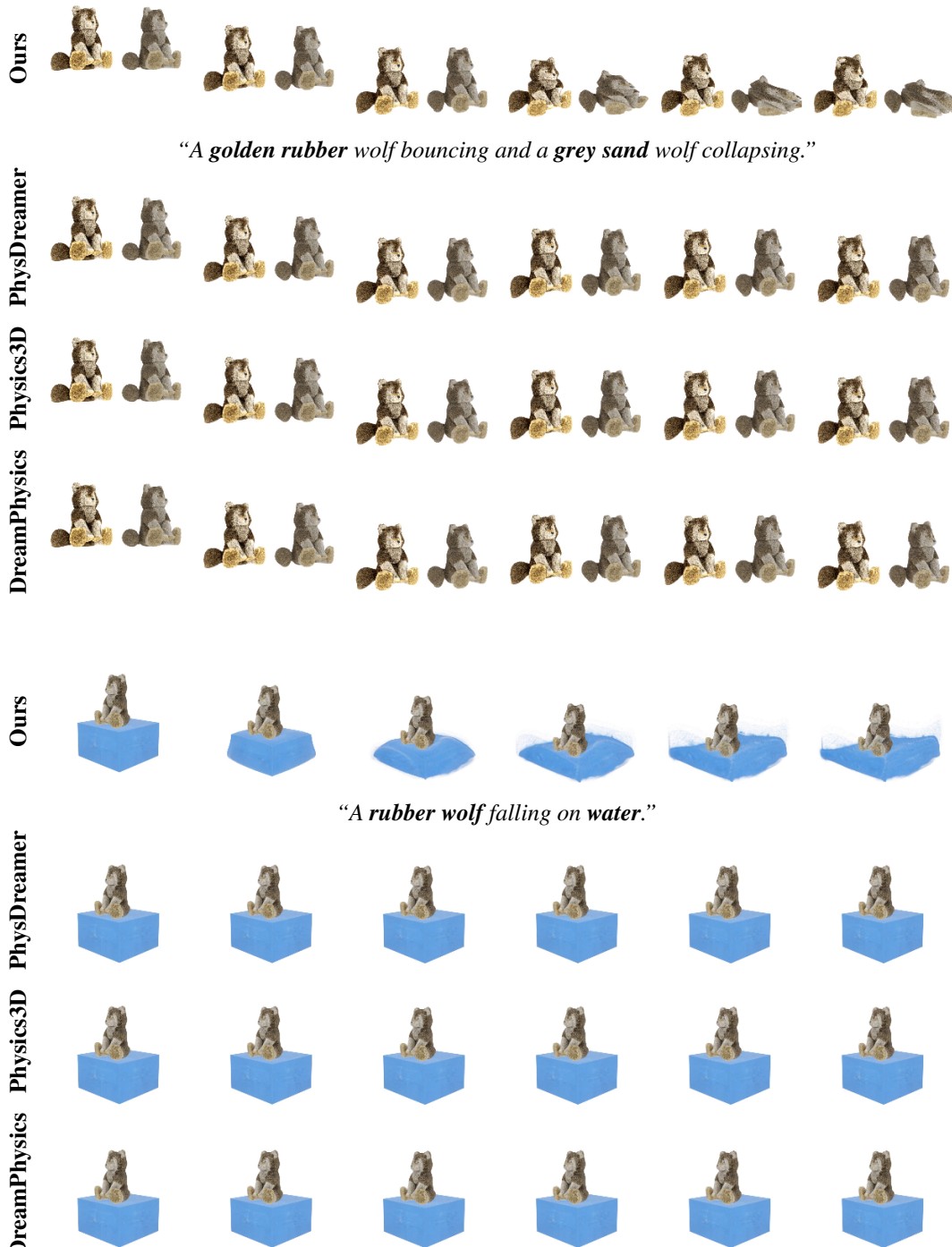

Figure 12: Qualitative visualizations of 3D dynamic synthesis for multiple objects in different materials. We present the results of our method and the baselines.

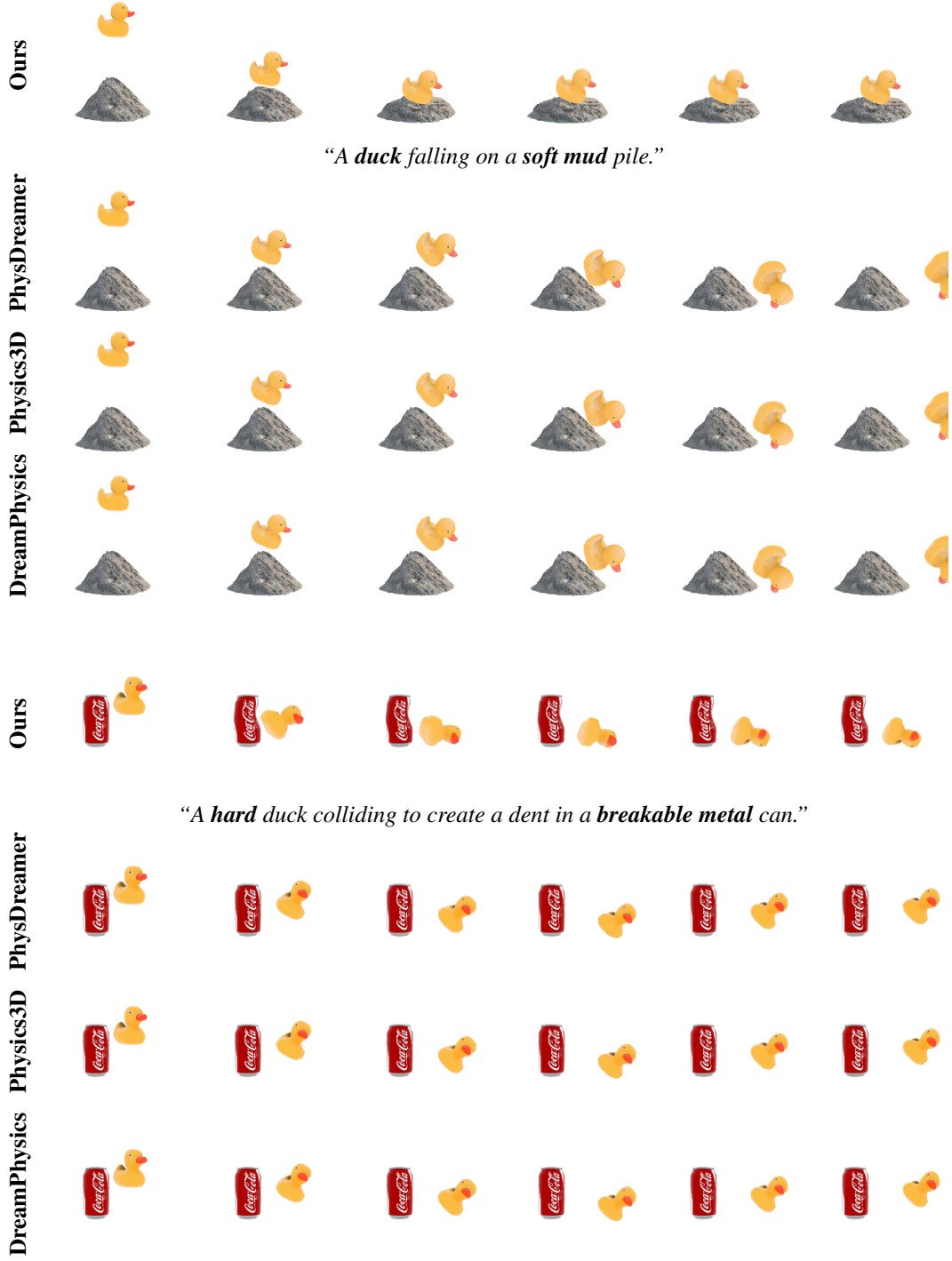

Figure 13: Qualitative visualizations of 3D dynamic synthesis for multiple objects in different materials. We present the results of our method and the baselines.

