# OpenReview forum: "OmniPhysGS: 3D Constitutive Gaussians for General Physics-Based Dynamics Generation"
_ICLR.cc/2025/Conference — ICLR 2025 Poster_

### Official Review · Reviewer_kAsd · 2024-10-26

**Soundness:** 3
**Presentation:** 3
**Contribution:** 3
**Rating:** 8
**Confidence:** 4

**Summary:**

The OmniPhysGS framework can simulate general physics-based 3D dynamics. It works with many types of materials like elastic, plastic, and fluid. Authors use Constitutive Gaussians and video diffusion models to make more realistic animations. It uses MPM for accurate physics simulation. The paper shows better results than PhysDreamer, Physics3D, and DreamPhysics.

**Strengths:**

This is the first method to use learnable constitutive models in Gaussian Splatting. It makes material simulation more flexible.

It supports many different materials. Other methods usually focus on one type.

The method uses text prompts and video diffusion models. It is very user-friendly and simple to use.

**Weaknesses:**

The quality depends on pre-trained diffusion models. This could make it difficult to simulate new materials or very specific materials.

The method might lack robustness when dealing with more complex or unusual physical scenarios, especially when the guidance models do not adequately capture the specific material properties.

**Questions:**

A discussion comparing the method to 'A generalized constitutive model for versatile MPM simulation and inverse learning with differentiable physics.' [2023] would be good. This would provide more context on how OmniPhysGS performs in comparison to recent advancements in MPM simulation and the capabilities of inverse learning. Authors should say more about the limits of the constitutive models. It is not clear what they assume about materials in each Gaussian particle. Would the proposed method be able to do material mixture as in this paper?

In experiments, it would be good to say how OmniPhysGS parameters are chosen. This is not clear for multi-object scenes where material differences are important.

More explanation needed about permanent deformation. How does OmniPhysGS handle it for viscoelastic or plastic materials? Would there be failure during training? For example, a bad parameter causes a 3D model to break down to pieces. The optmization may get stuck, right?

In Figure 3, could add some notes. Use a scene with multiple model categories, and show how the classification decoder assigns constitutive models to different areas of the scene.

The wolf on water scene, how is the shape of the water container specified in the pipeline?

---

> ### Author Response · Authors · 2024-11-16
> **Response to Reviewer kAsd (Part 1)**
>
> We sincerely appreciate the reviewer's insightful and valuable feedback.
> We are encouraged to know that you recognize the novelty of our method,
> which first introduces learnable constitutive models in Gaussian Splatting
> and supports various materials with user-friendly text prompts.
> We are really delighted that you like our work and lean towards acceptance.
> Below, we provide clarifications for the concerns raised.
> Additional analysis, experiments, and visualization results are included in the supplementary material,
> where we provide a Rebuttal Appendix (rebuttal-appendix.pdf) and supplementary videos (rebuttal-videos).  We greatly value your time and effort,
> and we welcome any follow-up questions or suggestions you may have.
>
> **1. The method may lack robustness when dealing with unusual materials or scenarios since the diffusion guidance may be limited.**
>
> Our method leverages the video diffusion model to guide the prediction of material properties. We acknowledge that the diffusion guidance may struggle with unusual corner cases and complex scenarios, which is a common limitation of data-driven methods.
> Despite the limitations, we claim that other core components of our method, such as the physics-guided network and the MPM simulator, can be optimized directly with the ground-truth motion of unusual materials to reconstruct the dynamics of the scene. This may mitigate the limitation of the diffusion guidance under unusual scenarios.
>
> **2. Discussion on the related work *A generalized constitutive model for versatile MPM simulation and inverse learning with differentiable physics* [1]. What are the limits of the constitutive models? Would the method allow for a material mixture?**
>
> We appreciate the reviewer's suggestion. In the related work [1], the authors propose a generalized constitutive model, which is a linear combination of several predefined constitutive models. The coefficients of the linear combination are learnable parameters. Therefore, the model can perform inverse learning to predict the coefficients from the observed motion. By adjusting the coefficients, they can simulate the materials mixture.
>
> The main difference and contribution, in terms of the generalized constitutive model, of our work compared to [2] and [3] are two-fold:
> - While [1] assumes a homogeneous scene (although mixed, the scene is still homogeneous based on a single mixed material),
> our method can handle **heterogeneous scenes with multiple objects and materials**, offering more flexibility.
> - Our method designs **a physics-guided network** to predict the material properties of each particle, which is more expressive than simply utilizing a set of coefficients. In easier tasks, such as inverse learning where the ground-truth motion is given, the coefficients can be optimized well as shown in [1].  However, our experiments in Section C of the Rebuttal Appendix prove that simply optimizing a probability vector (i.e., the coefficients) is hard to converge under our task setting.
>
> The constitutive model can capture various material properties, but they probably cannot perfectly conform to real-world physics. Meanwhile, off-the-shelf constitutive models are limited to several representative materials, which may not cover all the materials in the real world. Despite the limitations, we believe that the constitutive models can provide **a good approximation of real-world physics** and greatly enhance the physical plausibility of the generated dynamics.
>
> Our method can handle material mixture by simply removing the `argmax` operation in the physical-aware decoder and using the `softmax` output as linear coefficients for the constitutive models. However, in this work, we choose to assign a single material instead of a mixture of materials for each neighborhood. We claim that simply performing a linear combination of outputs of constitutive models may violate the original physical meaning of the constitutive models, which can lead to non-physical results and a lack of interpretability.
>
> We acknowledge that we may overlook some important works and would appreciate any suggestions.
>
> **3. How are the parameters being chosen?**
>
> We choose the simulator's hyperparameters such that the simulation is stable and the results are visually plausible. The specific choices of the parameters are based on our empirical experience without a large-scale search. Other hyperparameters, such as the learning rate and the number of training iterations, are also chosen based on our empirical experience. We find that the model is **not sensitive to these hyperparameters**.
>
> We initialize the weights of our neural network randomly (using the default initialization in PyTorch). The physical parameters of the particles are initialized to be the same for all particles in the scene. We choose the initial Young's modulus to be $2\times10^6$ and the Poisson's ratio to be $0.3$ according to NCLaw [2].

---

> ### Author Response · Authors · 2024-11-16
> **Response to Reviewer kAsd (Part 2)**
>
> **4. How does the method handle permanent deformation for viscoelastic and plastic materials?**
>
> The Material Point Method (MPM) can naturally handle permanent deformation for viscoelastic and plastic materials. Our method treats all deformations and materials in the same way.
>
> **5. Would there be a failure during training? For example, a bad parameter causes a 3D model to break down into pieces.**
>
> We understand the reviewer's concern. We addressed this issue by two strategies.
> - We adjusted the simulator's hyperparameters to ensure the stability of the simulation. Therefore, the model is less likely to generate physically unrealistic results. Notably, we only observe the breakdown or explosion of a 3D object when the simulating fails, such as when the simulating grids are too coarse or the time step is too large.
> - As mentioned in the manuscript, we utilize the multi-batch training strategy to deal with the optimization difficulty of MPM steps. Our method will optimize a stage multiple times from the same starting state. In this way, when non-physical results are generated in one stage, the diffusion guidance can correct them since the starting state is not changed.
>
> **6. More notes in Figure 3 can be added to illustrate how the decoder assigns different materials to different parts of the scene.**
>
> We appreciate the reviewer's suggestion. We have revised Figure 3 to include more notes to illustrate how the decoder assigns different materials to different parts of the scene. We presented the modified figure in Section D of the Rebuttal Appendix.
>
> **7. How is the shape of the water container specified in the wolf on the water scene?**
>
> We do not add any extra containers in that case.
> The reason why the water seems to be contained in a container is that
> the simulation is restricted to a $1\times1\times1$ cube.
> When initializing the particles,
> their positions are normalized to the range $[0, 1]$.
> During the simulation,
> we clamp the particles' positions to the range $[0, 1]$, which makes the water look like it is constrained in a container.
>
> We will add all the aforementioned discussions to the revision of this manuscript.
>
> ---
>
> [1] A Generalized Constitutive Model for Versatile MPM Simulation and Inverse Learning with Differentiable Physics. Proceedings of the ACM on Computer Graphics and Interactive Techniques (Symposium on Computer Animation), 2023.
>
> [2] Learning Neural Constitutive Laws From Motion Observations for Generalizable PDE Dynamics. The International Conference on Machine Learning (ICML), 2023.

---

> ### Author Response · Authors · 2024-11-25
> **A Kind Reminder**
>
> Thank you for taking the time to review our submission. We would like to kindly remind you to share any additional feedback or comments if possible. Your insights would greatly help us address any remaining concerns and further improve our work. We deeply appreciate your time and efforts.

---

### Official Review · Reviewer_S54K · 2024-11-04

**Soundness:** 3
**Presentation:** 3
**Contribution:** 3
**Rating:** 6
**Confidence:** 3

**Summary:**

In this work, the authors propose omniphysgs, a novel method for creating physics-based 3D dynamic scenes with diverse objects by modeling each asset as collections of 3D Gaussians and using multiple material sub-models. This method allows for complex material compositions, enhancing realism and flexibility in physical interactions. Experimental results show omniphysgs outperforms existing methods in visual quality and text alignment.

**Strengths:**

1. Performance: The proposed method omniphysgs achieves state-of-the-art results. The experiments well validate the effectiveness of the proposed methods.

2. Clarity: The paper is well-written and clearly structured, making the methodology and results easy to understand and follow.

3. Technical Novelty: The main contributions of this paper are twofold: 1) They propose a novel framework, which models each 3D asset as a collection of 3D Gaussians and represents physical materials using an ensemble of 12 domain-specific sub-models. This design significantly enhances the flexibility and realism of the synthesized dynamic scenes. 2) They define a scene by user-specified prompts and use a pretrained video diffusion model to supervise the estimation of material weighting factors, enabling the synthesis of more general and physically plausible interactions across a diverse range of materials.

**Weaknesses:**

Figure 1 is unclear regarding the material properties of the objects. It is confusing when the mountain is depicted as non-elastic while the duck toy is elastic, but the mountain collapses after the duck toy falls, which seems inconsistent with the assigned material properties.

**Questions:**

I am curious about how this method handles interactions between a single object and an entire scene, as opposed to interactions between two objects. Additionally, it would be better to understand the method's performance and behavior in scenarios involving more than two objects.

---

> ### Author Response · Authors · 2024-11-16
> **Response to Reviewer S54K**
>
> We sincerely appreciate the reviewer's insightful and valuable feedback.
> We are encouraged to know that you recognize the novelty and effectiveness of our method,
> and the breadth of the experiments,
> and that you found our manuscript well-written and easy to understand.
> We are truly delighted by your support and your inclination towards acceptance.
> Below, we provide clarifications for the concerns raised.
> Additional analysis, experiments, and visualization results are included in the supplementary material,
> where we provide a Rebuttal Appendix (rebuttal-appendix.pdf) and supplementary videos (rebuttal-videos).  We greatly value your time and effort,
> and we welcome any follow-up questions or suggestions you may have.
>
> **1. Figure 1 is confusing because the mountain is depicted as non-elastic but collapses after the duck falls.**
>
> We are sorry for the confusion caused by the example in Figure 1. The mountain is expected to be similar to a pile of plastic, deformable sand or mud. We depicted the mountain as non-elastic
> since its deformation is permanent and it does not recover its original shape after the deformation. In contrast, an elastic material would recover its original shape, such as the rubber duck in the same scene. Therefore, we think that the collapse of the mountain is consistent with the assigned material properties.
>
> **2. The interactions between a single object and an entire scene.**
>
> We appreciate the reviewer's suggestion.
> **We have conducted experiments on real-world scenes, including the flower vase scene and the fox scene.**
> Following PhysGaussian [1], a desired area of the scene is simulated and optimized to match the text prompt.
> We provide visualization results in Section E of the Rebuttal Appendix and supplementary videos. We hope that these experiments can demonstrate the effectiveness of our method in modeling the interactions between a single object and an entire scene.
>
> **3. The performance in scenarios involving more than two objects.**
>
> We appreciate the reviewer's suggestion.
> **We have conducted experiments on scenes involving more than two objects, including the pillow-basket scene and the material mixture scene.**
> Despite the memory-efficient MPM solver,
> the increased number of objects requires more GPU memory and computation, which may limit the complexity of the scene. Implementing a distributed version of the MPM solver is a potential future direction. Meanwhile, although the model can output desirable dynamics for each object in our experiments, we claim that it would be more challenging to perform fine-grained control over the interactions between multiple objects.
> We provide visualization results in Section E of the Rebuttal Appendix and supplementary videos.
>
> We will add all the aforementioned discussions to the revision of this manuscript.
>
> ---
>
> [1] PhysGaussian: Physics-integrated 3d Gaussians for generative dynamics. Proceedings of the IEEE/CVF Conference on Computer Vision and Pattern Recognition (CVPR), 2024.

---

> > ### Comment · Reviewer_S54K · 2024-11-25
> >
> > Thank you for your clarification and efforts in addressing the concerns. After reviewing the rebuttal materials and considering the other reviewers' comments, I have decided to maintain my original score but with a lower confidence level.

---

> > > ### Author Response · Authors · 2024-11-25
> > >
> > > Thank you for your thoughtful review and for taking the time to revisit our rebuttal materials. We understand and appreciate your decision to maintain your original score with a lower confidence level. If there are any additional points or clarifications that could further address your concerns, we would be more than happy to provide further explanations. Your feedback is invaluable in helping us refine our work.

---

### Official Review · Reviewer_ktNn · 2024-11-04

**Soundness:** 2
**Presentation:** 3
**Contribution:** 3
**Rating:** 6
**Confidence:** 5

**Summary:**

This paper focuses on generating dynamic 3D objects with physics-based simulation. The paper aims to relax the assumption of fixed constitutive models from previous works such as PhysGaussian and PhysDreamer. In particular, this paper introduces the OmniPhysGS framework with the learnable Constitutive Gaussians at the core. The main technical idea is to decompose an object into many small particle groups and use SDS optimization to assign a particular material to each group. The material assignment is a classification process that selects one of twelve predefined constitutive models with learnable physical parameters. Experiments are performed on a set of simple synthetic objects, showing that the proposed approach can generate different types of material dynamics.

**Strengths:**

- The problem is relevant and important. Current methods (e.g., PhysDreamer, Physics3D) are very much limited to elastic objects due to fixed constutive model. The proposed approach adds the flexibility of choosing from different materials.
- The experiment results of generating different types of material dynamics for a single input are interesting.
- Implementing a memory-efficient MPM solver is useful.

**Weaknesses:**

- There is no real experiments such as the ones in PhysDreamer and Physics3D.
- Some experiment results look weird, such as in 0:49 of the video, the can hit by the toy duck. It looks like the can has a very weird material (it does not look like metal at all), such that it keeps shrinking after being lightly hit by a toy.
- The metrics (e.g. CLIP similarity score) applying to videos do not seem to be very convincing to me. I'm not sure if CLIP score can be used to measure motion quality. I think the motion realism would be better judged by human preference.
- I'm a bit concerned about the technical design of dividing an object into small particle groups and allowing them to have different predefined constitutive models. That may not align with real-world physics. For example, it may give arbitrary material prediction in the interior of an object. I think the can example is one illustration of generating unreasonable results. Would this give reasonable results all the time?
- It is not clear to me why a neural network is needed at all. It seems there is no training, but per-scene test-time optimization. So there is no generalization issue. Then one may simply optimize a 12-way one-hot softmax vector for the material classification. What's the advantage of having to train a scene-specific neural network?
- Some notations are confusing, e.g., in L207, the physical parameters are denoted as a single real scalar value, yet the supplementary material says it includes Young’s modulus and Poisson’s ratio.

**Questions:**

Please see weaknesses above. I'm open to change my mind if there are evidences to rebut my points, though.

---

> ### Author Response · Authors · 2024-11-16
> **Response to Reviewer ktNn (Part 1)**
>
> We sincerely appreciate the reviewer's insightful and valuable feedback.
> We are encouraged to know that you recognize the significance of our target problem,
> as well as the flexibility and effectiveness of our method, which overcomes the limitations of existing approaches.
> We are also pleased that you found our memory-efficient MPM solver useful and the design of our experiments interesting.
> Below, we provide clarifications for the concerns raised.
> Additional analysis, experiments, and visualization results are included in the supplementary material,
> where we provide a Rebuttal Appendix (rebuttal-appendix.pdf) and supplementary videos (rebuttal-videos).  We greatly value your time and effort,
> and we welcome any follow-up questions or suggestions you may have.
>
> **1. No real-world experiments.**
>
> We appreciate the reviewer's suggestion.
> **We have conducted experiments on real-world datasets, including the flower vase scene and the fox scene.**
> We provide visualization results in Section E of the Rebuttal Appendix and supplementary videos.
> We hope that these experiments can demonstrate the effectiveness of our method in real-world scenarios.
>
> **2. The experiment result of the can-duck scene is weird.**
>
> We appreciate the reviewer's observation.
> By analyzing our collision experiments (e.g., the can-duck scene),
> we found that the unrealistic motion is caused by the **choice of grid resolution used in simulations**.
> The Material Point Method (MPM) utilizes grids to gather information from particles and subsequently transfer the information back to the particles.
> Consequently, using a lower grid resolution may lead to inadequate physical contact during collisions. This occurs because particles within a large grid may influence one another even when they are not in direct contact.
> In the original experiments, we used $25\times25\times25$ grids for all scenes for a fair comparison. Although this number is sufficient for most cases, low-resolution grids may cause artifacts such as collision without physical contact. Given the reviewer's feedback, we rerun the experiment with higher grid resolutions for the collision scene and achieve more realistic collision results.
> We provide analysis and visualization results in Section B of the Rebuttal Appendix and supplementary videos.

---

> ### Author Response · Authors · 2024-11-16
> **Response to Reviewer ktNn (Part 2)**
>
> **3. The CLIP metrics may be unconvincing. It's preferred to judge motion realism by human preference.**
>
> We appreciate the reviewer's suggestion. We agree that CLIP metrics may lack the ability to evaluate motion realism
> since the metrics are calculated on individual frames. As a complement to the metrics, we conducted a **user study among 20 participants** to evaluate the quality of the generated dynamics by different methods. During the study, the participants were asked to rank different videos based on both the text alignment and physical plausibility of the dynamics. The following table shows the detailed results of the user study, where the numbers represent the average ranking of each method. The lower the number, the better the performance.
>
> ### Single Object
>
> |              | Swinging Ficus | Collapsing Ficus |   Rubber Bear |   Sand Bear |    Jelly Cube |   Water Cube |   Average |
> |--------------|----------------|------------------|---------------|-------------|---------------|--------------|-----------|
> | PhysDreamer  |          3.158 |            3.125 |         2.059 |       2.769 |         2.176 |        2.538 |     2.638 |
> | Physics3D    |          2.158 |            2.750 |         2.882 |       3.000 |         2.647 |        3.000 |     2.740 |
> | DreamPhysics |          2.474 |            3.000 |         2.235 |       2.769 |         2.353 |        3.308 |     2.690 |
> | Ours         |          2.211 |            1.125 |         2.824 |       1.462 |         2.824 |        1.154 |     1.933 |
>
> ### Multiple Objects
>
> |              | Rubber and Sand | Duck and Pile | Rubber hits Metal | Bear into Water | Average |
> |--------------|-----------------|---------------|-------------------|-----------------|---------|
> | PhysDreamer  |           2.800 |         3.125 |             2.812 |           3.059 |   2.912 |
> | Physics3D    |           2.733 |         2.750 |             2.875 |           2.882 |   2.802 |
> | DreamPhysics |           3.067 |         2.688 |             3.062 |           3.059 |   2.935 |
> | Ours         |           1.400 |         1.438 |             1.250 |           1.000 |   1.351 |
>
> The results indicate that our method achieves better performance in modeling various kinds of materials. Specifically, the baselines achieve close performance to that of our method in modeling single pure elastic objects, but they struggle to model the behaviors of other materials (e.g., plasticity, viscoelasticity, fluid) especially when the scene is composed of multiple materials. This conclusion is consistent with the quantitative and qualitative results in Section 4.2 of the manuscript.
> We provide our user interface and better visualization of the table in Section A of the Rebuttal Appendix.

---

> ### Author Response · Authors · 2024-11-16
> **Response to Reviewer ktNn (Part 3)**
>
> **4. The technical design of dividing an object into small particle groups and allowing them to have different predefined constitutive models may not align with real-world physics.**
>
> The intuition behind our design is that **real-world scenes are usually composed of multiple materials.**
> Therefore, dividing the scene into small particle groups enables the model to assign different material properties to different parts of the scene according to the text prompt and the semantic or positional information of the particles. We understand the concerns when a single object is divided into multiple parts with different constitutive models, which may result in inconsistent behaviors of the object.
> During our experiments, we found that the overall physical properties of a single object are **determined by the majority of the particles**,
> which provides a reasonable approximation of real-world physics and enhances the flexibility of our method.
> We also find that our model can easily converge to a homogeneous material in a single-object scene (Section C of the Rebuttal Appendix).
>
> **5. Why a neural network is needed? What's the advantage of a neural network compared to optimizing a one-hot softmax vector for classification?**
>
> In our early experiments, we tried optimizing a vector representing the probability distribution over different constitutive models for each Gaussian particle. However, we found that this simple method was really **difficult to converge and prone to numerical instability**.
> In contrast, the neural network can effectively extract the features of the scene and utilize the neighborhood information of the particles to predict the material properties.
> We provide visualization comparison results of training our neural network and optimizing a one-hot softmax vector in Section C of the Rebuttal Appendix, where the neural network achieves better performance in terms of convergence and stability.
>
> **6. The notation of physical parameters.**
>
> We are sorry for the confusion caused by the notation of physical parameters. The physical parameter itself is a real scalar value, but a particle may have multiple kinds of physical parameters, such as Young's modulus and Poisson's ratio. Therefore, it would be better to use a vector to represent the physical parameters of a particle, i.e., $\boldsymbol{\gamma}\in\mathbb{R}^K$, where $K$ is the number of different physical parameters.
> We will revise the notation in the manuscript to make it more clear.
>
> We will add all the aforementioned discussions to the revision of this manuscript.

---

> ### Author Response · Authors · 2024-11-25
> **A Kind Reminder**
>
> Thank you for taking the time to review our submission. We would like to kindly remind you to share any additional feedback or comments if possible. Your insights would greatly help us address any remaining concerns and further improve our work. We deeply appreciate your time and efforts.

---

> > ### Comment · Reviewer_ktNn · 2024-11-25
> >
> > Thank the authors for the response.
> >
> > Regarding the points I raised:
> >
> > 1. Real examples. I think it is necessary to have a comparison with prior methods for the real examples. Only looking at the results of the proposed method is not very informative.
> >
> > Otherwise I'm fine with the rebuttal.
> >
> > I increased my score to 6. My overall evaluation is that this paper is not flawless (real experiments missing comparison, technical design not super convincing), but it has some merits that I believe the community might benefit from (optimizable constitutive model and efficient MPM implementation).

---

> > > ### Author Response · Authors · 2024-11-29
> > >
> > > Thank you for your thoughtful feedback and for revisiting our work.
> > > We greatly appreciate your willingness to raise your score.
> > >
> > > Since we cannot edit the supplementary materials now,
> > > we will include more comparisons with baseline methods for real examples in the final version of our manuscript
> > > and provide a project website for visualization and further details.
> > >
> > > If you have any further questions or require additional information, we are more than happy to provide any clarification or details you may need.

---

### Official Review · Reviewer_h6D1 · 2024-11-04

**Soundness:** 2
**Presentation:** 2
**Contribution:** 3
**Rating:** 6
**Confidence:** 4

**Summary:**

This paper proposes a physics-based dynamics generation method that includes different physical material properties. The authors model the object as constitutive 3D Gaussians that exhibit multiple physical material properties by an ensemble of physical domain-expert sub-models. The dynamics can be described and input as a user prompt, the learnable constitutive models would be optimized by a pre-trained video diffusion model with SDS loss. The authors designed a 3D feature extractor and physical-aware decoder to model ordinary Gaussians to constitutive Gaussians. Due to the limited generation length of existing video diffusion models, the authors designed two training strategies, grouping and multiple mini-batch training to allow the training on MPM simulation steps.

**Strengths:**

1. This approach uses 3D Gaussians as constitutive models, enabling the representation of diverse physical material properties. It facilitates the optimization of these properties via a learnable MPM simulation, which integrates video SDS loss.

2. Domain-expert constitutive models are incorporated to steer the learning process across various materials, functioning similarly to a Mixture of Experts methodology.

3.  Two training strategies are introduced, grouping and multi-mini-batch training, to address the challenges posed by numerous MPM simulation steps and the limited number of frames produced by video diffusion models,

**Weaknesses:**

1. The authors limit their comparisons to the same object modeled with vastly different materials. This approach is less persuasive because altering materials in PhysGaussian [1] is straightforward, whereas optimizing appropriate physical parameters is challenging. More convincing evidence would come from showing varied parameter results within the same material. I will elaborate on this point in the questions section.

2. The collision experiments appear to lack physical contact, consistently showing a gap between the colliding objects, whereas collisions modeled in PhysGaussian [1] are depicted as more substantial. The overall visual quality of the experiments still falls short of what is achieved with PhysGaussian.

**Questions:**

The experiments primarily compare the same object using two distinctly different materials, such as rubber versus sand or jelly versus fluids. However, in practical applications, it is unusual to utilize the same object with radically different materials. Moreover, modifying the material type in PhysGaussian [1] is relatively straightforward, requiring changes to only three fields in the configuration file: the material itself, Young’s modulus, and Poisson’s ratio. This simplicity contrasts sharply with the complexity of optimizing a model using video diffusion models for similar tasks. From my perspective, it is beneficial to explore the optimization of physical parameters for a single type of material. For instance, metals like gold exhibit high plasticity, whereas aluminum alloys display low plasticity. Similarly, wood varieties may vary significantly in flexibility. As demonstrated in Figure 6 of PhysGaussian [1], jelly can show varying degrees of stiffness and volume preservation by adjusting Young's modulus and Poisson's ratios. These parameters are challenging to adjust manually and merit further exploration for optimization via video diffusion models.

As stated in line 234 by the authors, “Unlike previous works (Zhang et al., 2024b; Liu et al., 2024; Huang et al., 2024) that maintained fixed constitutive models, Constitutive Gaussians allow the model to capture diverse material behaviors, encompassing both elastic and plastic deformations, thus offering a more dynamic and comprehensive representation of material properties.” Additionally, Equation 4 highlights that the hyperelastic energy density function, the plasticity return function, and the physical parameters are all learnable parameters within the constitutive models. Therefore, as the authors assert, this approach should facilitate the optimization of material properties within the same material category.

I am eager to see experimental results that demonstrate this capability of the model. For instance, comparing a "hard rubber bear" with a "stretchy rubber bear," or a "hard metal can" with a "soft metal can" would illustrate the model’s effectiveness. Presenting such results would prompt me to raise my score.

[1] Xie, T., Zong, Z., Qiu, Y., Li, X., Feng, Y., Yang, Y., & Jiang, C. (2024). Physgaussian: Physics-integrated 3d Gaussians for generative dynamics. In Proceedings of the IEEE/CVF Conference on Computer Vision and Pattern Recognition (pp. 4389-4398).

---

> ### Author Response · Authors · 2024-11-16
> **Response to Reviewer h6D1**
>
> We sincerely appreciate the reviewer's insightful and valuable feedback.
> We are encouraged to know that you appreciate our work,
> which incorporates 3D Constitutive Gaussians,
> integrates domain-expert constitutive models,
> and employs effective training strategies for learning material properties in dynamic generation.
> Below, we provide clarifications for the concerns raised.
> Additional analysis, experiments, and visualization results are included in the supplementary material,
> where we provide a Rebuttal Appendix (rebuttal-appendix.pdf) and supplementary videos (rebuttal-videos).
> We greatly value your time and effort,
> and we welcome any follow-up questions or suggestions you may have.
>
> **1. The comparisons are limited to the same object with different materials. More experiments on controlling the physical parameters of the same object are needed.**
>
> We appreciate the reviewer's suggestion.
> Our method is capable of controlling the physical parameters of the same object.
> To demonstrate the flexibility of our method in controlling the physical strength,
> such as the softness and hardness, of the same object,
> **we conducted experiments including the ficus scene, the wolf scene, the jelly scene, and the material mixture scene.**
> We provide visualization results in Section E of the Rebuttal Appendix and supplementary videos.
>
> **2. Altering materials in PhysGaussian [1] is straightforward. This simplicity contrasts sharply with the complexity of optimizing a model using video diffusion models for similar tasks.**
>
> We agree that altering materials in PhysGaussian [1] is straightforward. This simplicity is because their experiments are limited to a single, homogeneous object/scene. In contrast, our method is designed to **handle more complex scenarios**, such as scenes with multiple objects and different materials.
> In this case, assigning appropriate material properties to each object is challenging and our method provides a flexible and effective solution.
>
> **3. The collision experiments appear to lack physical contact, whereas collisions modeled in PhysGaussian [1] are depicted as more substantial.**
>
> We appreciate the reviewer's observation. By analyzing our collision experiments (e.g., the can-duck scene), we found that the unrealistic motion is caused by the **choice of grid resolution used in simulations**. The Material Point Method (MPM) utilizes grids to gather information from particles and subsequently transfer the information back to the particles. Consequently, using a lower grid resolution may lead to inadequate physical contact during collisions. This occurs because particles within a large grid may influence one another even when they are not in direct contact.
> In the original experiments, we used $25\times25\times25$ grids for all scenes for a fair comparison. Although this number is sufficient for most cases, low-resolution grids may cause artifacts such as collision without physical contact.
> Given the reviewer's feedback, we rerun the experiment with higher grid resolutions for the collision scene and achieve more realistic collision results.
> We provide analysis and visualization results in Section B of the Rebuttal Appendix and supplementary videos.
>
> We will add all the aforementioned discussions to the revision of this manuscript.
>
> ---
> [1] PhysGaussian: Physics-integrated 3d Gaussians for generative dynamics. Proceedings of the IEEE/CVF Conference on Computer Vision and Pattern Recognition (CVPR), 2024.

---

> ### Author Response · Authors · 2024-11-25
> **A Kind Reminder**
>
> Thank you for taking the time to review our submission. We would like to kindly remind you to share any additional feedback or comments if possible. Your insights would greatly help us address any remaining concerns and further improve our work. We deeply appreciate your time and efforts.

---

> ### Comment · Reviewer_h6D1 · 2024-11-26
>
> I still have doubts about the second point, that even for multiple objects, assigning material types is not complex. And the examples are not actually complex scenes in my view.
> Also, I agree with review ktNn that there are flaws in the paper and more convincing results are needed.
> However, since the authors provided additional results for the different parameters for the same material properties and resolved the collision problem, and considering the pytorch implementation of MPM, I raised my score to 6.

---

> > ### Author Response · Authors · 2024-11-29
> >
> > Thank you for taking the time to revisit our rebuttal and for providing detailed feedback.
> > We appreciate your willingness to raise your score.
> >
> > Regarding the second point,
> > our motivation is to provide a flexible framework that can handle scenarios with arbitrary material types and material parameters,
> > while prior methods fail to do so.
> > Since we cannot edit the supplementary materials now,
> > we will include more comparisons with baseline methods to demonstrate the flexibility of our approach in the final version of our manuscript
> > and provide a project website for visualization and further details.
> >
> > If you have any further questions or require additional information, we are more than happy to provide any clarification or details you may need.

---

### Official Review · Reviewer_7Xoc · 2024-11-04

**Soundness:** 3
**Presentation:** 3
**Contribution:** 2
**Rating:** 6
**Confidence:** 5

**Summary:**

The paper presents a framework for text-driven physical parameter learning for 3DGS-reconstructed objects. Compared to previous methods, the main novelty lies in making discrete constitutive model types learnable based on given text prompts. For each particle, the physics-aware decoder predicts a material type from 12 predefined material classes. The simulated and rendered results are fed into a pretrained text-to-video diffusion model to evaluate SDS for guiding both material class and material parameter learning. Efficiency enhancements are also proposed, including optimizing the long sequence in chunks.

**Strengths:**

- The overall presentation of the framework is clear.
- Comparisons and ablations are extensive.

**Weaknesses:**

- The controls of the scene—such as initial velocity and external forces—are not learnable parameters. For this reason, the paper is best summarized as a framework for learning physical parameters, where interactions between objects remain fixed and are not directly controllable. Consequently, text descriptions must be crafted based on these known, predefined interactions.
- In reality, objects cannot be both elastic and sandy. The physical plausibility is questionable. With such heuristic blending of materials within a single object, the dynamics may conform to unrealistic motions encoded in the video model that is impossible in reality.

**Questions:**

- How is the physical-aware decoder trained? The material classes are discrete and the sampling of $j_i, k_i$ in Eq.6 is non-differentiable. How does the model learn to change material class? This is the key novelty of the paper, a detailed discussion is needed.

- The following references are highly related, which also design learnable constitutive models:
    - Nagasawa, Kentaro, et al. "Mixing sauces: a viscosity blending model for shear thinning fluids." ACM Trans. Graph. 38.4 (2019): 95-1.
    - Su, Haozhe, et al. "A generalized constitutive model for versatile mpm simulation and inverse learning with differentiable physics." Proceedings of the ACM on Computer Graphics and Interactive Techniques 6.3 (2023): 1-20.


- Warp can create arrays from Torch tensor without copying. I am wondering why it consumes much more memory than implementing MPM in Pytorch.

---

> ### Author Response · Authors · 2024-11-16
> **Response to Reviewer 7Xoc (Part 1)**
>
> We sincerely appreciate the reviewer's insightful and valuable feedback.
> We are encouraged to know that you recognize
> the novelty of our proposed learnable Constitutive Gaussians,
> the thoroughness of our comparisons and ablations,
> and that you find this work well-presented.
> Below, we provide clarifications for the concerns raised.
> Additional analysis, experiments, and visualization results are included in the supplementary material,
> where we provide a Rebuttal Appendix (rebuttal-appendix.pdf) and supplementary videos (rebuttal-videos).
> We greatly value your time and effort,
> and we welcome any follow-up questions or suggestions you may have.
>
> **1. How is the physical-aware decoder trained given that the discrete material class is non-differentiable?**
>
> The physical-aware decoder is trained using a differentiable approximation of the discrete material class.
> Specifically, the direct output of the physical-aware decoder is a continuous vector of shape `(num_particles, num_materials)`,
> which is then passed through a `softmax` function to obtain a probability distribution over the materials.
> We choose the material with the highest probability for each particle, i.e., `argmax(softmax(output))`.
> Since the `argmax` operation is non-differentiable,
> its gradient is estimated using the  **straight-through estimator**,
> which is a common trick in training nondifferentiable operations like `argmax`.
> The whole process is defined in the following differentiable `hard_softmax` function:
> ```
> def hard_softmax(logits: Tensor, dim: int) -> Tensor:
>     y_soft = logits.softmax(dim=dim)
>     index = y_soft.argmax(dim=dim, keepdim=True)
>     y_hard = torch.zeros_like(y_soft).scatter_(dim=dim, index=index, value=1.0)
>     ret = y_hard - y_soft.detach() + y_soft
>     return ret
> ```
> In the forward pass, `hard_softmax` behaves the same as an `argmax` operation, while in the backward pass, it behaves the same as a `softmax` operation.
>
> **2. The control of the scene, such as initial velocity and external forces, is not learnable parameters.
> The text descriptions must be crafted based on these known, predefined interactions.**
>
> In this work, we mainly focus on **learning the material properties** of a given scene, which includes constitutive models and physical parameters such as Young's modulus. This task is challenging because it is difficult to assign appropriate material properties to numerous particles, especially in scenes that involve multiple materials. The interactions between objects are implicitly governed by these material properties.
>
> We do not learn the initial velocity and external forces
> since users can easily control these factors (e.g., by dragging an arrow in an interactive user interface).
> It is noteworthy that the simulator-based feature of our method allows generalizing the learned material properties to new scenes with different initial conditions, for which we have conducted experiments in Section 4.2 Motion Generalization of the manuscript.
>
> In terms of text descriptions,
> although it would be better to describe the motion,
> only describing the material also works in our method since **the material implicitly determines the dynamics**.
> For example, the prompt ``a rubber bear'' implies the bear is bouncy and elastic.
> New experiments are provided in Section E of the Rebuttal Appendix, where input text descriptions do not contain any verbs.
>
> **3. In reality, objects cannot be both elastic and sandy. Given such heuristic blending of materials within a single object, the dynamics may conform to unrealistic motions encoded in the video diffusion model.**
>
> We apologize for the confusion caused by the ficus tree example in Figure 4,
> which can appear either elastic or sandy depending on the prompts used.
> However, this demonstration highlights the flexibility in generating diverse material properties
> and illustrates how text prompts can be used to **control these materials effectively**.
> We believe there are some objects that may look quite similar
> but have different material properties,
> such as a blue jelly cube and a water cube with the same shape (Figure 7 in the manuscript appendix).
> In this case, the material properties can be controlled by the text prompt.
>
> We admit the ambiguity and potential artifacts introduced by the data-driven nature of the video diffusion model. Our method predicts material properties using the conditional probability distributions provided by this model. Since the video diffusion model is trained on large real-world datasets, it is expected to assign higher probabilities to material properties that are more realistic and text-consistent, thus enhancing the physical plausibility of our method.

---

> ### Author Response · Authors · 2024-11-16
> **Response to Reviewer 7Xoc (Part 2)**
>
> **4. Warp can create arrays from Torch tensor without copying. Why does Warp consume much more memory than implementing MPM in Pytorch?**
>
> Although WARP can create arrays from PyTorch tensors without copying,
> **it automatically creates zero-gradient arrays and other intermediate arrays** when the gradient is required.
> These additional arrays are not managed by PyTorch and can consume a substantial amount of extra memory due to the numerous MPM simulation steps involved. Removing these intermediate arrays is non-trivial, requiring significant modifications to the WARP library according to our early experiments. Therefore, previous work like PhysDreamer [1] had to employ KMeans clustering to reduce the number of particles used in simulations.
>
> Our implementation of MPM in PyTorch takes advantage of the advanced memory management tools available in the framework, such as half-precision training and gradient checkpointing. In this work, we utilize gradient checkpointing as an effective compromise between memory usage and computational efficiency. Additionally, our implementation minimizes the communication overhead between PyTorch and WARP, resulting in reduced testing time. We hope that our PyTorch-based MPM implementation will serve as a valuable tool for future research.
>
> **5. More related works.**
>
> We appreciate the reviewer's suggestion. In the related work,
> *Mixing Sauces: A Viscosity Blending Model for Shear Thinning Fluids* [2]
> and *A Generalized Constitutive Model for Versatile MPM Simulation and Inverse Learning with Differentiable Physics* [3],
> the authors propose a generalized constitutive model.
> Specifically, [2] proposes to blend different constitutive models in a non-linear way and [3] proposes a linear combination of several predefined constitutive models.
> The weights or coefficients of each predefined constitutive model are learnable parameters. Therefore, the model can perform inverse learning to predict the coefficients from the observed motion.
>
> The main difference and contribution, in terms of the generalized constitutive model, of our work compared to [2] and [3] are two-fold:
> - While [2] and [3] assume a homogeneous scene (although mixed, the scene is still homogeneous based on a single mixed material),
> our method can handle **heterogeneous scenes with multiple objects and materials**, offering more flexibility.
> - Our method designs **a physics-guided network** to predict the material properties of each particle, which is more expressive than simply utilizing a set of coefficients. In easier tasks, such as inverse learning where the ground-truth motion is given, the coefficients can be optimized well as shown in [2] and [3].
> However, our experiments in Section C of the Rebuttal Appendix prove that simply optimizing a probability vector (i.e., the coefficients) is hard to converge under our task setting.
>
> We acknowledge that we may overlook some important works and would appreciate any suggestions.
>
> We will add all the aforementioned discussions to the revision of this manuscript.
>
> ---
>
> [1] PhysDreamer: Physics-based interaction with 3d objects via video generation. European Conference on Computer Vision (ECCV), 2024.
>
> [2] Mixing Sauces: A Viscosity Blending Model for Shear Thinning Fluids. ACM Transactions on Graphics (TOG), 2019.
>
> [3] A Generalized Constitutive Model for Versatile MPM Simulation and Inverse Learning with Differentiable Physics. Proceedings of the ACM on Computer Graphics and Interactive Techniques (Symposium on Computer Animation), 2023.

---

> ### Author Response · Authors · 2024-11-25
> **A Kind Reminder**
>
> Thank you for taking the time to review our submission. We would like to kindly remind you to share any additional feedback or comments if possible. Your insights would greatly help us address any remaining concerns and further improve our work. We deeply appreciate your time and efforts.

---

> > ### Comment · Reviewer_7Xoc · 2024-11-26
> >
> > Thank you for addressing my questions. After considering the rebuttal and other reviews, I am willing to increase the score to 6.

---

> > > ### Author Response · Authors · 2024-11-29
> > >
> > > Thank you for your thoughtful consideration and for revisiting our rebuttal and the other reviews.
> > > We greatly appreciate your willingness to adjust your score.
> > > If you have any further questions or require additional information, we are more than happy to provide any clarification or details you may need.

---

### Author Response · Authors · 2024-11-16
**General response**

We would like to sincerely thank all the reviewers for their valuable, constructive, and thoughtful feedback. It is really inspiring to know that the majority of the reviewers consider that

(1) the proposed method is novel, meaningful, and effective, which introduces learnable Constitutive Gaussians, a physics-guided network, and two efficient training strategies for dynamic generation;

(2) the engineering design is well-organized, including the implementation of a memory-efficient MPM solver, and extensive quantitative and qualitative experiments;

(3) the manuscript is well-written and easy to follow.

We address each of the reviewer's comments in detail in the individual responses. Additionally, further analysis, experiments, and visualization results are included in the supplementary material. A Rebuttal Appendix (rebuttal-appendix.pdf) and supplementary videos (rebuttal-videos)
are provided in the zip file, without modifying the original manuscript.

We sincerely hope that our responses are helpful and informative. If there are still any concerns that we have not addressed, we would greatly appreciate any further feedback and are more than willing to make improvements where necessary.

Thank you again for your time and valuable input!

---

### Meta-Review · Area_Chair_QEnx · 2024-12-21

**Metareview:**

The paper introduces OmniPhysGS, a framework for generating 3D dynamic scenes with diverse and realistic material behaviors.  Reviewers generally found the method novel and the paper well-written, but some questioned unrealistic collisions and other aspects of physical plausibility.  Concerns were also raised about the lack of complex real-world experiments and the need for more detailed comparisons with existing methods.

Reviews were ultimately unanimously positive, and thus the paper is accepted.

**Additional Comments On Reviewer Discussion:**

Scores were initially mixed, with some borderline negative. Reviewers initially expressed concerns about the lack of real-world experiments, unrealistic collisions, limited comparisons, technical design plausibility, reliance on pre-trained models, and motion quality evaluation. The authors responded by adding real-world experiments, re-running collision experiments, expanding comparisons, and clarifying design choices. Despite a few persisting doubts, two reviewers raised their scores, and the reviews now remain unanimously positive.

---

### Decision · Program_Chairs · 2025-01-22

Accept (Poster)